# ELI trifocal microscope: a precise system to prepare target cryo-lamellae for in situ cryo-ET study

Shuoguo Li [1,2,4], Ziyan Wang[2,3,4], Xing Jia[1,2], Tongxin Niu[1], Jianguo Zhang[1], Guoliang Yin[2,3], Xiaoyun Zhang[1], Yun Zhu [2,3] ✉, Gang Ji [1,2] ✉ & Fei Sun [1,2,3] ✉

Cryo-electron tomography (cryo-ET) has become a powerful approach to study the high-resolution structure of cellular macromolecular machines in situ. However, the current correlative cryo-fluorescence and electron microscopy lacks sufficient accuracy and efficiency to precisely prepare cryo-lamellae of target locations for subsequent cryo-ET. Here we describe a precise cryogenic fabrication system, ELI-TriScope, which sets electron (E), light (L) and ion (I) beams at the same focal point to achieve accurate and efficient preparation of a target cryo-lamella. ELI-TriScope uses a commercial dual-beam scanning electron microscope modified to incorporate a cryo-holder-based transfer system and embed an optical imaging system just underneath the vitrified specimen. Cryo-focused ion beam milling can be accurately navigated by monitoring the real-time fluorescence signal of the target molecule. Using ELI-TriScope, we prepared a batch of cryo-lamellae of HeLa cells targeting the centrosome with a success rate of ~91% and discovered new in situ structural features of the human centrosome by cryo-ET.

Three-dimensional (3D) visualization of the cellular ultrastructure is an important step in understanding life. With rapid technology development, cryo-electron microscopy (cryo-EM) single-particle analysis has become one of the most important tools to study 3D high-resolution structures of biomacromolecules in vitro[1]. Meanwhile, cryo-ET has also been rapidly developed, becoming a unique technique to study the in situ high-resolution structures of biomacromolecular complexes and the locations and interactions of these complexes with their native cellular environment[2], which is poised to bring another revolutionary breakthrough in structural biology in the near future[3,4].

However, many obstacles still exist in applying cryo-ET widely and efficiently for in situ structural studies, especially the low quality, low efficiency and low accuracy of specimen preparation methods[5,6]. Due to the limited penetration distance of electrons at the current accelerating voltage (300 kV) of the modern microscope, a thin (~200 nm) cryo-section of a cell or tissue specimen should be prepared. Cryo-focused ion beam (FIB) (cryo-FIB) milling has been proven to be an efficient method to prepare high-quality cryo-lamellae of cells for in situ structural study[7,8], with many successful applications[9–11]. The lamella can be milled to a thickness of ~50–300 nm without the conventional artefacts caused by cryo-ultramicrotomy[12]. Recently, 3D architectures of various cellular organelles, such as the cytoskeleton[10], the endoplasmic reticulum[13] and the 26S proteasome[14] have been revealed in situ by cryo-FIB and cryo-ET.

Furthermore, to enable accurate preparation of cryo-lamellae in the target region by cryo-FIB milling, cryo-correlative light and electron microscopy (cryo-CLEM) has been developed. The target region in the cell can be labeled with a fluorescent molecular probe (for example, green fluorescent protein (GFP)) and imaged by various cryo-fluorescence microscopies (FMs) (cryo-FMs). Next, the

[1]Center for Biological Imaging, Core Facilities for Protein Science, Institute of Biophysics, Chinese Academy of Sciences, Beijing, China. [2]University of Chinese Academy of Sciences, Beijing, China. [3]National Key Laboratory of Biomacromolecules, CAS Center for Excellence in Biomacromolecules, Institute of Biophysics, Chinese Academy of Sciences, Beijing, China. [4]These authors contributed equally: Shuoguo Li, Ziyan Wang. ✉e-mail: zhuyun@ibp.ac.cn; jigang@ibp.ac.cn; feisun@ibp.ac.cn

fluorescence image can be used to navigate cryo-FIB fabrication. The conventional cryo-CLEM workflow requires a stand-alone fluorescence microscope with a cryo-stage for cryo-fluorescence imaging[6,15,16], followed by cryo-scanning electron microscopy (SEM) (cryo-SEM). A correlation alignment between the cryo-FM and cryo-SEM images is then generated and used to guide cryo-FIB milling[17–19]. This workflow is complicated, and ensuring sample quality is challenging due to multiple transfers between microscopes[20,21]. Each transfer increases the risk of sample devitrification and ice contamination, leading to reduced accuracy of the correlation alignment[22,23]. Moreover, specific fiducial markers imaged by both cryo-FM and cryo-FIB, such as fluorescent beads, are required for the correlation alignment using specific 3D correlative software[20]. Accurate z-axis positions of both fiducial markers and fluorescent targets are necessary for precise correlation; however, this information can only be roughly determined by widefield FM[23], spinning-disk confocal microscopy[20] or cryo-Airyscan confocal microscopy[17] with essentially limited resolution[18,24,25].

Recently, another cryo-CLEM concept has been developed by integrating a fluorescence imaging system into the cryo-SEM chamber to avoid specimen transfer during microscopy[26], and this is used in commercially available products including iFLM[27] (Thermo Fisher Scientific) and METEOR[28] (Delmic). Using these integrated systems, cryo-FM images can be acquired before and after cryo-FIB milling to check the presence of the target signal without increasing the risk of contamination. However, these current systems potentially have a correlation resolution limitation due to the low numerical aperture (NA) of the objective lens, and the cryo-FIB milling efficiency is limited due to the frequent switching between cryo-FM and cryo-FIB.

Here, starting from the concept of our previously developed high-vacuum optical platform for cryo-CLEM (HOPE)[15], we developed a new cryo-CLEM system named ELI-TriScope to achieve accurate and efficient preparation of target cryo-lamellae. Based on a commercial dual-beam SEM, ELI-TriScope incorporates a cryo-holder-based transfer system and embeds an inverted fluorescence imaging system just underneath the vitrified specimen. In ELI-TriScope, an electron (E) beam, a light (L) beam and an ion (I) beam are precisely adjusted to the same focal point; as a result, cryo-FIB milling can be accurately navigated by monitoring the real-time fluorescence signal of the target molecule. With ELI-TriScope, there is no need to add fiducial markers or perform sophisticated correlation alignment between cryo-FM and cryo-SEM, leading to markedly improved efficiency, accuracy, success rate and throughput of cryo-FIB milling compared with other reported cryo-CLEM techniques.

To evaluate the efficiency of ELI-TriScope, the human centriole was selected as a challenging target to perform an in situ structural study. A batch of cryo-lamellae of HeLa cells targeting the centrosome were efficiently prepared with a success rate of ~91%. The subsequent cryo-ET study not only confirmed the ultrastructure of various typical components in human centrioles but also revealed new structural features. Therefore, ELI-TriScope provides a highly successful solution for sample preparation for in situ cryo-ET study and will have wide application in future in situ structural biology.

## Results
### Design of the ELI-TriScope system
Our ELI-TriScope system is developed based on a dual-beam SEM and contains two major components, a custom-designed cryo-holder-based vacuum transfer system for SEM imaging and an inserted fluorescence imaging system just underneath the vitrified specimen inside the SEM chamber (Fig. 1a, Extended Data Fig. 1 and Supplementary Video 1).

For the vacuum transfer system, the original chamber door of the SEM is replaced by a new concave door equipped with an outside three-axis piezo (New Focus 9161M, Newport) and servomotor (Maxon) stage tightly mounted with a cryo-holder adaptor (Extended Data Fig. 1a,b). The three-axis stage allows high-precision movement

in three dimensions. A worm wheel mechanism is designed to drive the three-axis stage to tilt in the range of −70 to 55 degrees by the servomotor (Maxon) with a precision of 0.01 degrees. A homemade airlock prepump vacuum system developed previously[15] connects the cryo-holder adaptor and the concave door (Fig. 1a). The cryo-holder can be loaded into the SEM chamber via the adaptor and through the prepump vacuum system. Next, the position of the cryo-holder can be adjusted by controlling the three-axis stage, which allows precise localization of the cryo-specimen at the crossover point of the electron and ion beams. The movement and tilt of the stage can be controlled via customized software written in LabVIEW 2011 (National Instruments).

Inside the SEM chamber, a widefield optical imaging system is inserted just underneath the tip of the cryo-holder (Fig. 1a, Extended Data Fig. 1a–c and Methods). We named this optical imaging system the cryogenic SimulTAneous monitoR (cryo-STAR) system. To maximize the detection sensitivity for weak fluorescence signals of target molecules in real time, a high-NA dry objective lens is selected and installed in this system. Meanwhile, an epifluorescence system with a full-spectrum (4,6-diamidino-2-phenylindole (DAPI)−GFP−red fluorescent protein (RFP)−Cy5) white light-emitting diode (LED) light source is equipped. Fluorescence signals from the vitrified specimen are collected by the objective lens and recorded by a high-sensitivity complementary metal−oxide−semiconductor (CMOS) camera.

The ELI-TriScope system adjusts the electron beam, ion beam and light beam to the same focal point. Therefore, the fluorescence signal of target molecules can be monitored in real time while cryo-FIB milling is being performed (Supplementary Video 1). As a result, the cryo-FIB fabrication procedure can be accurately navigated to the specific region of interest (Fig. 1b). Because there is no need for specimen transfer, the risk of ice contamination, specimen damage and devitrification can be largely avoided, and the operation time of fabricating one cryo-lamella can be efficiently reduced.

### ELI-TriScope workflow
The centrosome is a highly ordered organelle that controls cell proliferation, motility, signaling and architecture[29]. As the microtubule (MT)-organizing center of vertebrate cells, centrosomes organize the formation of two poles of the mitotic spindle during cell division and act as templates for the formation of flagella and cilia[30,31]. Each centrosome in a cell consists of an MT-based cylindrical component named the centriole and surrounding pericentriolar material[32]. Most of the time, there are only two centrioles closely localized in one cell, and these two centrioles are termed the mother and daughter according to their maturity degrees. Therefore, preparing a cryo-lamella containing centrioles for in situ structural study is very difficult using a conventional cryo-FIB procedure without precise fluorescence signal navigation. We therefore select the human centriole as a challenging target to validate the performance of our ELI-TriScope technique with the following workflow (Fig. 2 and Supplementary Video 2).

First, we selected HeLa cells expressing mCherry fluorescent protein-labeled pericentrin, which is an integral component of the centrosome located in the pericentriolar material region[33]. The fluorescence signal of mCherry fluorescent protein was used to navigate cryo-FIB milling to the target centrosome. The fluorescence-labeled cells were seeded onto a custom-designed ultraviolet-sterilized OD grid (O-shaped grid with D-shaped mesh) and subjected to cryo-vitrification by plunge freezing. Next, the cryo-vitrified grid was assembled into an FEI AutoGrid and further loaded onto a commercial cryo-EM multiholder (Gatan), ready for the subsequent cryo-transfer (Fig. 2a). Notably, there is one straight bar in the OD grid that can be used as an orientation indicator. During loading of the grid onto the cryo-holder, we adjusted the orientation of the grid to ensure that this straight bar was parallel to the holder tilt axis and therefore perpendicular to the subsequent cryo-FIB milling direction. Next, in the subsequent cryo-ET

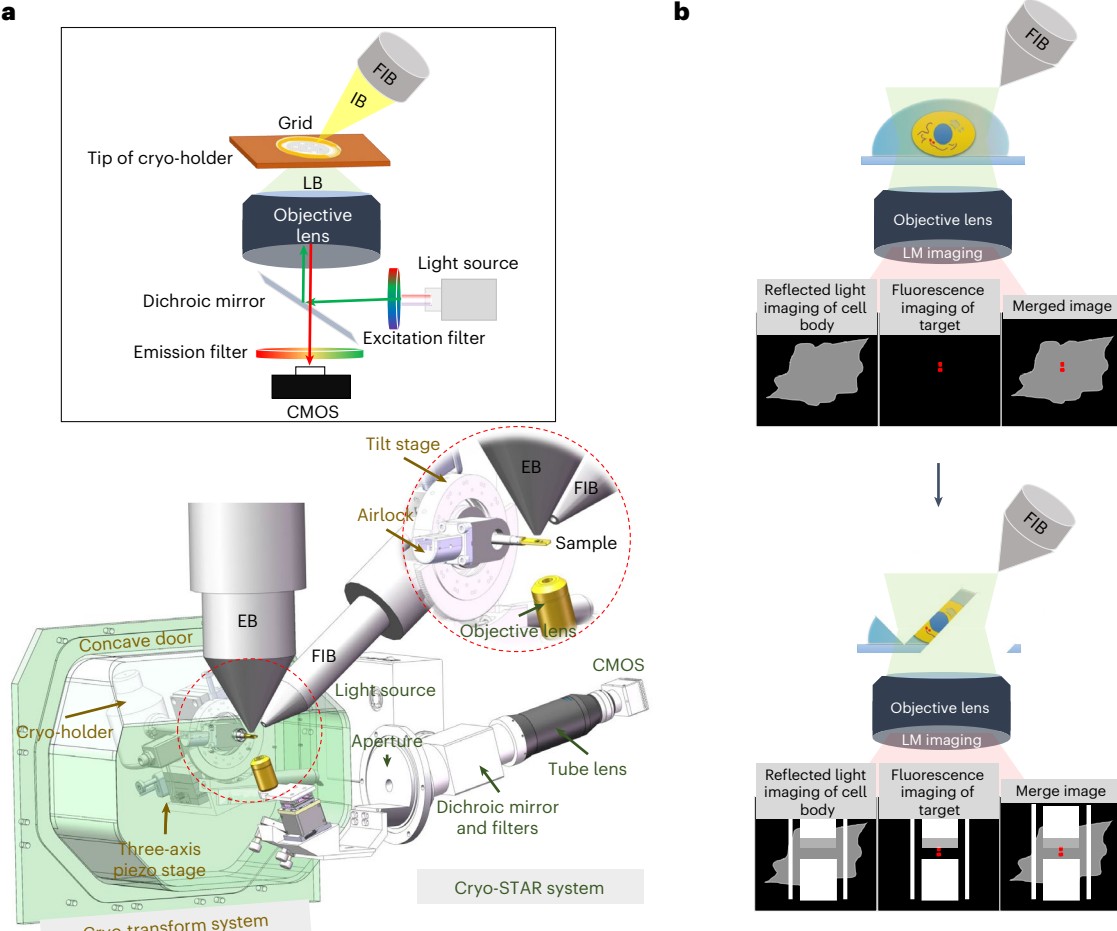

**Fig. 1 | Design and principle of ELI-TriScope. a**, Schematic diagram and design drawing of ELI-TriScope in its operational mode. Each part of the system is labeled and described. IB, ion beam; LB, light beam; EB, electron beam. **b**, Working principle of ELI-TriScope. The ion beam and light beam are simultaneously focused on the area of interest of the cryo-specimen. The reflected light image recorded by the camera displays the cell body, and the target signal in the fluorescence image is used to navigate cryo-FIB milling. LM, light microscope.

data collection, we loaded the AutoGrid and adjusted its orientation to ensure that this bar was parallel to the tilt axis.

Second and optionally, we used our previously developed cryo-FM HOPE system[15] or the HOPE structured illumination FM system[34] to screen the cryo-vitrified cells (Fig. 2b). Both bright-field and fluorescence images under cryogenic conditions were recorded to check the thickness of the specimen and the intensity of fluorescence signals. The coordinates of the selected cells were recorded onto the map of the grid, which was used in the subsequent experiment. This step can be skipped if we can ensure that the specimen is well vitrified and contains well-distributed fluorescence signals. Notably, the cryo-transfer between our HOPE system and the subsequent ELI-TriScope system uses the AutoGrid to protect the EM grid, minimizing the risk of specimen deformation, devitrification and ice contamination.

Third, the cryo-holder with the cryo-vitrified specimen was loaded into the ELI-TriScope system. At the beginning, the cryo-holder was tilted to 30 degrees to allow the cryo-grid to face the gas injection system (GIS) (Extended Data Fig. 1c), and the predefined positions on the grid were coated with a protective organometallic layer (Extended Data Fig. 1d). Next, the holder was tilted back to −20 degrees, allowing the objective lens of the cryo-STAR system to rise to the focus position (Extended Data Fig. 1e).

Fourth, FIB and FM images were acquired with magnifications of 2,500 and 100, respectively (Fig. 2c). The target position was identified according to the fluorescence signals in the FM image with the reference coordinates recorded in the above cryo-FM screening (optional). Next, the stage was controlled to move the target position to the crossover point of the ion and electron beams, and the position of the objective lens of the cryo-STAR system could be further optimized for the best focus. Both FIB and FM images were acquired again. With good precalibration of the ELI-TriScope system (Methods), the fluorescence signal (two adjacent bright spots for two centrioles near the cell nucleus) in the FM image can be well matched to the cell feature in the FIB image. Next, the coarse milling process was performed with a large beam current of ~0.23–0.43 nA around the target position, monitored by real-time fluorescence imaging with the cryo-STAR system. The drift of the specimen was detected in the fluorescence image and corrected by precisely controlling the stage. When the thickness of the cryo-lamella was less than 3 μm, fine milling was performed with a smaller beam current of ~40–80 pA.

During the milling process, the fluorescence signal was always kept in the center of the cryo-lamella, and the milling position was adjusted accordingly. At the beginning of milling, the two centrioles showed two strong adjacent fluorescence peaks with their intensities accurately monitored (Fig. 2c). The start of the decrease in their intensity indicated that the milling reached the target. Next, the milling process should be stopped and started again on another side of the cryo-lamella. Finally, the cryo-lamella was trimmed to less than 200 nm in thickness by maintaining a sufficiently strong fluorescence

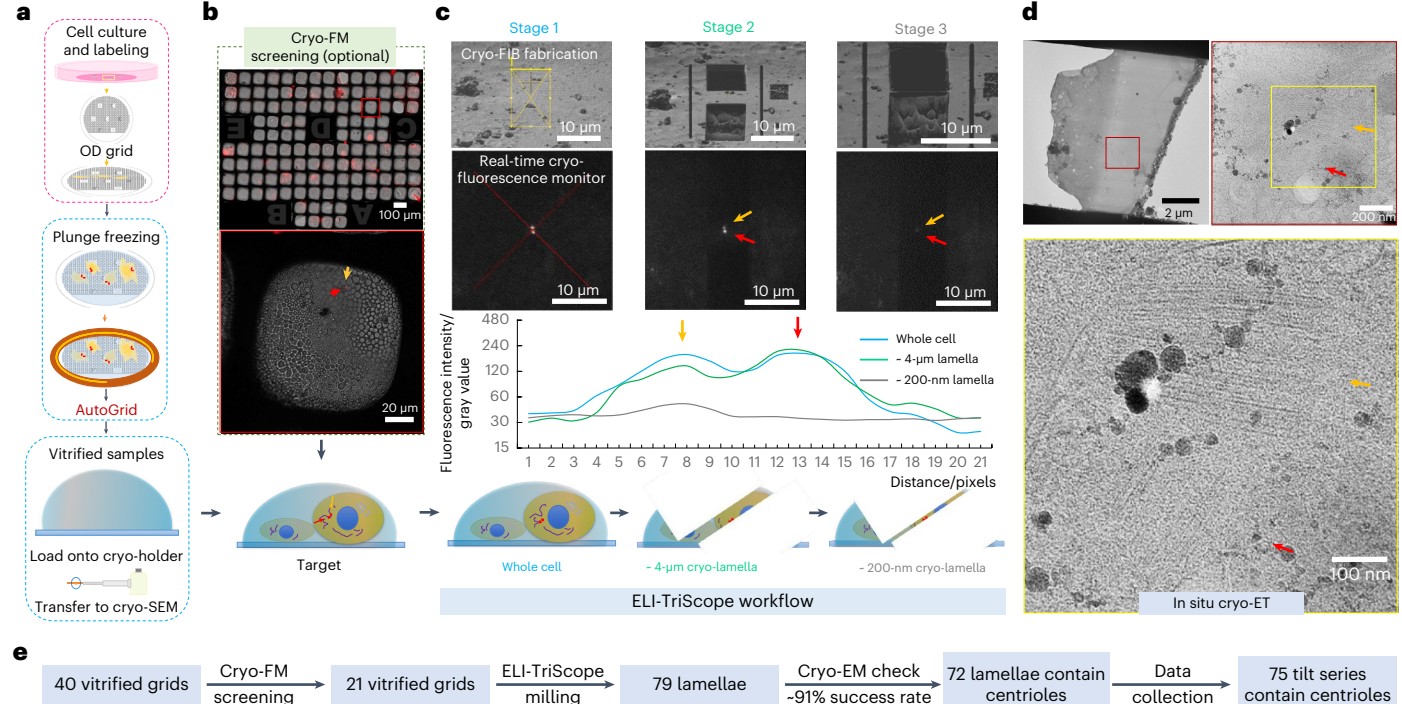

**Fig. 2 | Cryo-CLEM workflow using ELI-TriScope. a**, Cells labeled by fluorophores are cultured on an OD grid (T11012SS, TIANLD), loaded into an FEI AutoGrid (Thermo Fisher Scientific), vitrified by plunge freezing and then loaded onto a cryo-holder. **b**, Screening of cryo-vitrified cells using cryo-FM (optional). The ice thickness and the fluorescence signal are checked at this step. The bright-field atlas merged with fluorescence images of the grid can be acquired, and the positions of cells with good fluorescence signals (orange and red arrows) are recorded. **c**, Real-time fluorescence signal-navigated cryo-FIB milling by ELI-TriScope. During the cryo-FIB process, the cryo-fluorescence intensity of the target is monitored in real time. In the SEM and fluorescence images from left to right, three different stages during cryo-FIB milling are presented: the beginning stage (left), the middle stage with one side rough milled (middle) and the final stage with two sides fine milled (right). The fluorescence signals from two centrioles are indicated by red and orange arrows and monitored in real time. A line profile of the intensity of the fluorescence signal is correspondingly plotted below for the three stages. A schematic diagram of the thickness of the cryo-lamella at different stages is shown at the bottom. **d**, The prepared cryo-lamella is transferred into a cryo-transmission electron microscope (TEM) (cryo-TEM) for cryo-ET data collection. Top, cryo-EM micrographs of the cryo-lamella at low (×3,600, left) and high (×33,000, right) magnifications. Bottom, a zoomed-in view of the cryo-EM micrograph at high magnification to show the centrioles, which are labeled with orange and red arrows. **e**, Statistics of our whole cryo-CLEM workflow from grid vitrification to cryo-ET tilt series data collection.

intensity. Because the centriole has an overall size larger than 200 nm, the final fluorescence intensity decayed in comparison with the original fluorescence intensity.

After precise cryo-FIB fabrication, the cryo-lamella containing the target centrioles can be transferred to a cryo-electron microscope for subsequent cryo-ET data collection and in situ structural study (Fig. 2d).

In comparison with other reported cryo-CLEM techniques, including both nonintegrated[29–31] and integrated[26–28,35] workflows, ELI-TriScope simplifies the specimen-transfer steps and reduces the time cost to prepare one cryo-lamella from ~2–2.5 h per cryo-lamella to ~0.8 h per lamella (Extended Data Fig. 2).

In addition, the precision and success rate of ELI-TriScope are also substantially improved. In this study, we prepared 40 vitrified cryo-grids, and 21 of them showed good quality (proper ice thickness and well-distributed fluorescent signals) in the cryo-FM screening and were used for ELI-TriScope milling. We fabricated a total of 79 cryo-lamellae and found 72 of them containing centrioles, which were used for the subsequent cryo-ET data collection (Fig. 2e, Supplementary Fig. 1 and Supplementary Data 1). As a result, based on our current statistics, we calculated the success rate of targeting centrioles by using ELI-TriScope as ~91%.

### In situ structure of the human centriole

Human centrioles are closely related to tumorigenesis and multiple hereditary diseases[36]. Revealing the centriole-assembly details and

the roles of each centrosomal component is important. The centrioles are of variable size among species, while, in mammalian cells, typical mature centrioles are approximately 230 nm in diameter and 500 nm in length[37]. Nine sets of MT triplets (MTTs) are arranged in ninefold symmetry, and each MTT contains a full MT (A-tubule) and two partial MTs (B-tubule and C-tubule). Centrioles are polarized along their longitudinal axis, possessing different structures at the proximal and distal ends. At the proximal end, the cartwheel acts as a determinant scaffold for centriole symmetry[38] but is degraded in mature human centrioles[39]. At the distal end, there are distal appendages (DAs) for mediating the attachment of ciliary vesicles to centrioles during ciliogenesis and subdistal appendages (SDAs) for positioning centrioles and cilia by anchoring MTTs[40,41].

From 72 cryo-lamellae, we collected 75 tilt series containing centrioles (Fig. 2e). Next, we selected 46 well-aligned tilt series and 55 centrioles with high integrity for further structural analysis. The average thickness of our processed cryo-lamellae was 169 ± 43 nm, which is smaller than the narrowest part of the human centriole (Extended Data Fig. 3a). Therefore, in each tomogram, only partial centrioles exist. In some tomogram slices, we observed two newborn procentrioles next to the mature centrioles, indicating that these cells should be in the S phase of the cell cycle (S[1]) and that the centrosomes are under replicating conditions (Extended Data Fig. 3c,d).

Both top and side views of centrioles could be observed, and no obvious preferred orientations were found in the current dataset

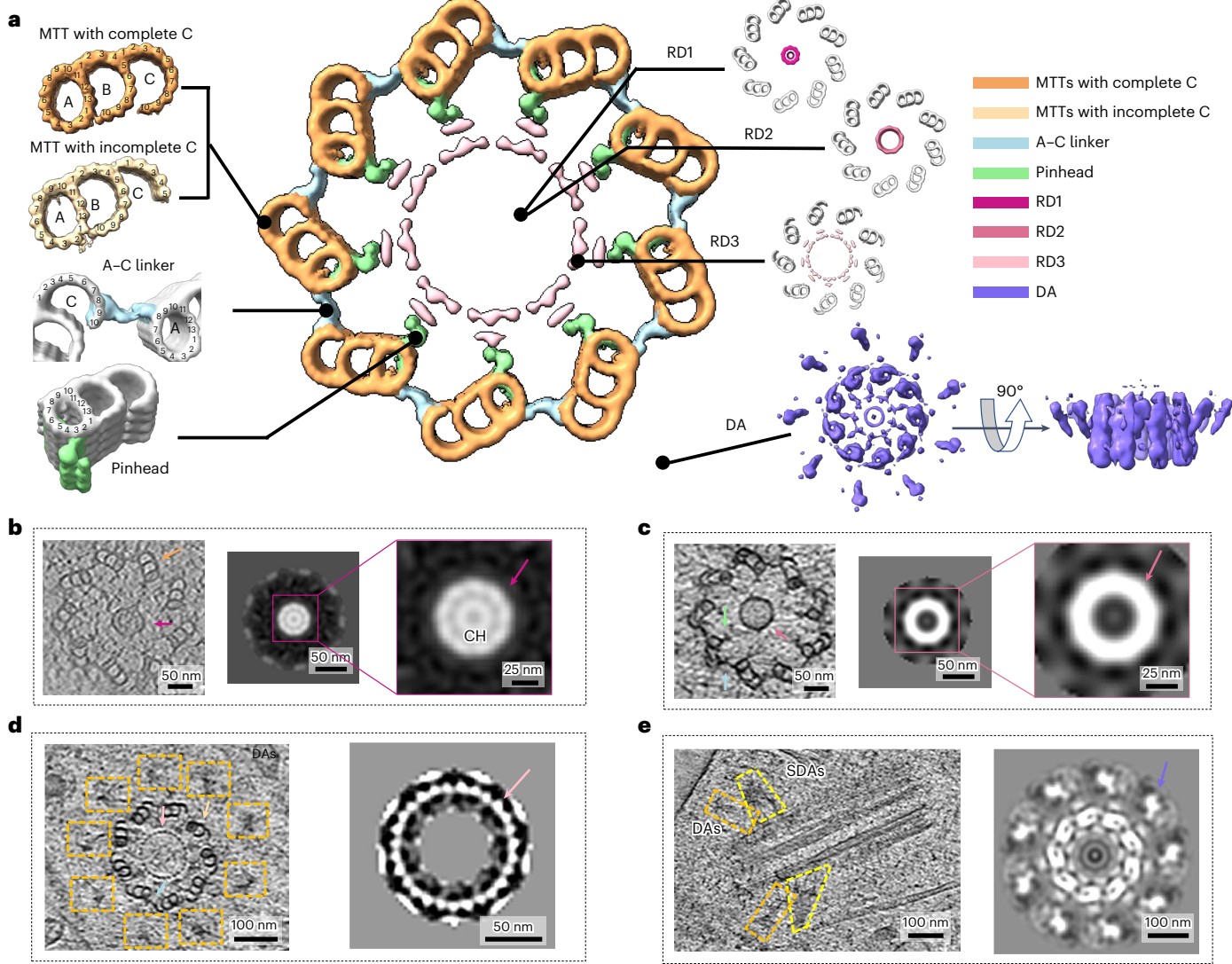

**Fig. 3 | In situ structure of the human centriole. a**, Cross-sectional view of human centriole slices in different local regions, which are reconstructed by STA. MTTs with a complete C-tubule and an incomplete C-tubule are colored sandy brown and Navajo white, respectively. The A–C linker, pinhead, RD1, RD2, RD3 and DA are colored light blue, light green, medium violet red, pale violet red, pink and medium slate blue, respectively. **b**, RD1 and MTTs with a complete C-tubule indicated by the arrows in the tomogram (left) and a cross-sectional view of the RD1 map (ninefold symmetry applied) obtained by STA (right). The RD1 peak is indicated by the arrow. The CH is observed inside. The tomogram slice shows one representative RD1 structure selected from three observed ones. **c**, RD2, A–C linker and pinhead indicated by the arrows in the tomogram (left) and cross-sectional view of the RD2 map (ninefold symmetry applied) obtained by STA (right).

The RD2 peak is indicated by the arrow. Only one representative RD2 structure was observed in the dataset. **d**, RD3, MTTs with an incomplete C-tubule, A–C linker and DA indicated by arrows and boxes in the tomogram (left) and cross-sectional view of the RD3 map (ninefold symmetry applied) obtained by STA (right). The 27 rod-like density peaks of RD3 are indicated by the arrow. The tomogram slice shows one representative RD3 structure selected from three observed ones. **e**, SDA and DA indicated by boxes in the tomogram (left) and cross-sectional view of the DA map (ninefold symmetry applied) obtained by STA (right). The tomogram slice shows one centriole with representative SDA and DA structures selected from two observed ones. The baseball bat-shaped density of the DA is indicated by the arrow. The color schemes in **b**–**e** are the same as that in **a**.

(Extended Data Fig. 3b), which avoids anisotropy reconstruction in the following data processing. All subvolumes of centrioles were cropped from the original tomograms along the axis from the proximal end to the distal end at 4-nm intervals and then aligned to different local regions using the subtomogram averaging (STA) approach (Extended Data Fig. 4 and Extended Data Table 1).

The human centriole shows a regular ninefold symmetry structure in both the tomogram slice and the STA reconstruction (Fig. 3 and Supplementary Video 3), which was reported to be due to the strict regulation of the highly conserved spindle-assembly abnormal protein 6 (SAS-6)[31,37–39]. Almost all the walls of centrioles are formed by MTTs, but, in the very distal end of the centriole, MT doublets (MTDs) (A-tubules

and B-tubules) are observed, while the C-tubule vanishes (Extended Data Fig. 5). For the MTTs, according to the unbiased data processing and 3D classifications, approximately three-quarters of MTTs have a complete C-tubule containing ten protofilaments and sharing three protofilaments (fifth, sixth and seventh) with the B-tubule, while a quarter of the MTTs have an incomplete C-tubule containing only approximately five protofilaments (Fig. 3a and Extended Data Fig. 6). Based on the relative positions of these two elements, MTTs with a complete C-tubule are located nearer the proximal end of the centriole, and those with an incomplete C-tubule are closer to the distal end (Extended Data Fig. 7), which is consistent with previous reports[42].

The diameter of the centriole distal end where MTDs emerge is ~220 nm (Extended Data Fig. 8a), while the regions in the same centriole where MTTs with an incomplete C-tubule exist have an average diameter of ~240 nm (Extended Data Fig. 8b). When MTTs with a complete C-tubule exist, centrioles show a larger diameter of ~270 nm (Extended Data Fig. 8c). Our observation that the human centriole has the narrowest diameter in the distal region is consistent with previous studies[37,43].

According to the STA reconstructions of MTTs and single A-tubules (~25 Å resolution), the MT walls of the human centriole have a repeating periodicity of ~8.4 nm (Extended Data Fig. 6), similar to that in *Paramecium tetraurelia* (*P. tetraurelia*) and *Chlamydomonas reinhardtii* (*C. reinhardtii*) and centrioles isolated from *Homo sapiens* (*H. sapiens*) KE37 cells[44]. For MTTs with a complete C-tubule, a linker density can be observed between the A-tubules and C-tubules from adjacent MTTs, referred to as the A–C linker (Fig. 3). In the STA reconstructions of the A–C linker, we found that it binds to the eighth protofilament of the A-tubule in one MTT and the eighth and ninth protofilaments of the C-tubule in the adjacent MTT (Fig. 3a), consistent with previous reports on *Trichonympha agilis* and Chinese hamster ovary cells[37,45]. This shows that the A–C linker conformation is highly conserved among species. For those MTTs with an incomplete C-tubule, the A–C linker density is missing in some regions but remains between the A-tubule and the edge of the incomplete C-tubule (Fig. 3c,d). It seems that, even for an incomplete C-tubule with only approximately five protofilaments, the A–C linker can still connect it to the A-tubule of the adjacent MTT. Possibly, the A–C linker connects with other protofilaments in the incomplete C-tubule.

In addition to the A–C linker, another important component that interacts with the MTT is the pinhead. The pinhead has been reported to extend from the MT wall to bridge the A-tubule and the central cartwheel in the proximal end of the centriole[46,47]. However, whether the pinhead exists in the mature centriole of humans remains unknown. In this study, we observed the pinhead density in most centrioles (Fig. 3c). It contains at least two fibrous components and interacts with the third protofilament of the A-tubule (Fig. 3a). The remaining part of the pinhead that links the central cartwheel is highly dynamic; it can only be traced in some tomogram slices but is averaged out in the STA reconstruction.

### Internal and external structures of human centrioles

Precise assembly of the centriole and maintenance of cohesion stability require internal scaffold structures. The cartwheel was the first observed internal structure and believed to be the scaffold for centriole biogenesis as well as the determinant for the centriole symmetry, which has been studied in several species, such as *P. tetraurelia* and *C. reinhardtii*[38,39,46,48]. In the human centriole, we observed several types of ring structures with different shapes and diameters inside MTTs (Fig. 3b–d). The unambiguous top-view particles corresponding to three major populations (Fig. 3b–d), named ring density (RD)1, RD2 and RD3, were manually picked and aligned separately (Extended Data Fig. 4). We applied ninefold symmetry during alignment and averaging, yielding three ring-like maps with diameters of approximately 50 nm, 65 nm and 100 nm for RD1, RD2 and RD3, respectively (Fig. 3a). The overall shapes of these maps appear consistent with their original densities in the raw tomograms (Fig. 3b–d).

Central hubs (CHs) (~25 nm in diameter) were reported to exist in the middle of centrioles in the proximal region[39]. In this study, we found similar structures in RD1 from the STA map and some raw tomograms (Fig. 3a,b) but not in all the RD1-existing centrioles. RD1 is probably a cartwheel-related structure, but its internal components may change during the development stage of centrioles. Unlike the reported longitudinal periodicity of ~4.2 nm along the CH from isolated centrioles of *H. sapiens*[39], we did not find similar periodicity of RD1 in our STA analysis.

The diameter of RD2 is slightly larger than that of RD1, and no structural densities were found within RD2 (Fig. 3a), which is consistent with the raw tomogram (Fig. 3c). The blurred density of RD2 suggests that RD2 might have different symmetry than ninefold or have high dynamics.

RD3 is another internal structure of the human centriole observed in our tomograms, and, to the best of our knowledge, it has not been discussed before. It has a diameter of 100 nm and contains 27 peak densities along the wall (Fig. 3d and Supplementary Video 3). It has no obvious inner structure and is located near the pinhead region. Interestingly, this RD3 structure is often associated with MTTs possessing an incomplete C-tubule and a DA (Fig. 3d and Supplementary Video 3), indicating that RD3 is located at the distal end of the centriole and may be related to DA components. In the STA volume with ninefold symmetry, RD3 shows 27 clear rod-like structures, suggesting that it may play a role in maintaining the cohesion of MTTs in the distal end region. Meanwhile, we noted an RD3-like density in a previous study of centrioles using the plastic section of HeLa cells[40]; however, it was not carefully and explicitly studied.

DA is a typical structure of mature centrioles in many species, and it is also observed in our tomogram slice of the human centriole (Fig. 3e and Supplementary Video 4). Each DA protrusion connects to one MTT and extends toward the outward and distal ends of centrioles, the terminus of which has a diameter of ~450 nm. In the STA volume with ninefold symmetry, the DA shows a baseball bat-shaped structure of each protrusion (Fig. 3a). Interestingly, in this STA volume, we also observed a clear RD3 density 100-nm in diameter that connects to the pinhead of MTTs inside the centriole. For the SDA, it can be found near the DA in the tomogram slice (Fig. 3e and Supplementary Video 4) but is averaged out in the STA reconstruction.

## Discussion

To achieve precise cryo-FIB fabrication of cells and efficiently prepare target cryo-lamellae for subsequent cryo-ET in situ structural study, we developed an ELI-TriScope system by incorporating a cryo-STAR system into a commercial dual-beam SEM. To enable the electron beam, ion beam and light beam to be focused on the same coincidence point, a new cryo-transfer system using a commercial cryo-holder was designed to replace the conventional chamber door of the dual-beam SEM, which allows the objective lens of cryo-STAR to be positioned just underneath the cryo-specimen. Our cryo-transfer system uses the AutoGrid to protect the EM grid, and touching the EM grid with the specimen can be largely avoided during the whole cryo-CLEM workflow. The reduced number of specimen transfers largely decreases the risk of specimen damage, deformation, devitrification and ice contamination.

In the ELI-TriScope system, after careful precalibration, the electron beam, ion beam and light beam are focused on the same point, and then the cryo-specimen can be imaged simultaneously by the FIB beam and the light beam with the same field of view. Therefore, cryo-FIB milling can be precisely navigated by monitoring the real-time fluorescence signal, resulting in accurate cryo-fabrication of cells in the target region. Compared to previously reported cryo-CLEM workflows for cryo-lamella preparation, our ELI-TriScope technique does not require pre-spreading of fluorescence beads in the sample, avoids the sophisticated correlation procedure between cryo-FM and cryo-FIB images and simplifies the overall workflow of site-specific cryo-lamella preparation with a high success rate.

In the present work, we applied our ELI-TriScope technique to study the in situ structure of the human centriole in HeLa cells. Recent advances in cryo-EM have greatly promoted the structural study of centrioles in *C. reinhardtii*, *P. tetraurelia*, *Naegleria gruberi*, *H. sapiens*, etc.[37,39,44]. However, great difficulties are still encountered in the study of high-resolution structures of centrioles in their native environment. For *C. reinhardtii*, the exact location of the centriole can be tracked along the flagella, but the location of centrioles in most other species

cannot be easily targeted. This leads to a substantially lower efficiency of cryo-FIB milling, which in turn greatly limits the in situ structural study of centrioles.

Using ELI-TriScope, we were pleased to observe that we could prepare 72 high-quality cryo-lamellae of HeLa cells with a high-success target rate. We were therefore able to collect a large amount of cryo-ET tilt series of cryo-lamellae and study the in situ structure of the human centriole. We observed multiple components of human centrioles in their native states. With a preliminary STA procedure, we could resolve the single protofilaments of MTTs and observe their longitudinal periodicity.

With the ninefold geometry in the centriole of almost all organisms, the MTs that comprise the centriole cylinder can be singlet, doublet or triplet, depending on species. It has been illustrated that centrioles in early *Caenorhabditis elegans* embryos have MT singlet assembly[49]. In addition, centrioles from *Drosophila melanogaster* S2 cells are composed of a mixture of singlets and doublets[37], whereas those in sperm cells are uniquely long and consist of MTTs[50]. In addition, previous studies showed that some species, including *C. reinhardtii* and human KE37 cells, have MTTs in the proximal region and MTDs in the distal ends[44]. Here, we found that human centrioles in HeLa cells are mostly composed of MTTs and only have MTDs at the very distal end region. Moreover, we also observed many internal and external densities of human centrioles, including RD1, RD2, RD3 and DAs. Besides aligning with previous reports, the existence of RD1–RD3 in the human centriole has not been reported before. We found that RD1 is associated with the cartwheel in the proximal region and RD3 is associated with the DA and is located around the distal region. In addition, a helical inner scaffold structure was found to maintain MTT cohesion under compressive force[44], and it can also be traced in the distal regions of centrioles in our dataset (Supplementary Video 3). These discoveries of ours will be further studied by collecting more cryo-ET data and improving the completeness and resolution of STA in the future.

Overall, we developed an advanced precise and efficient cryo-FIB fabrication technique with the name ELI-TriScope and applied this technique to study the 3D in situ structure of the human centriole. By further improving resolution and sensitivity of the cryogenic fluorescence module, ELI-TriScope will have wide application in future in situ structural biology and in studying the high-resolution ultrastructure of specific events in the cell.

## Online content

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

## Methods

### Incorporating the cryo-STAR system into a dual-beam SEM

To build the cryo-STAR system (Fig. 1a), an embedded widefield optical imaging system was mounted onto an existing port of a commercial dual-beam SEM (FEI Helios 600i, Thermo Fisher Scientific). The light path occupies a large port, and the control cables use a small port. In the vacuum chamber, a high-NA dry objective lens (LMRlanFL N ×100; NA/WD, 0.8/3 mm; Olympus) was placed underneath the vitrified specimen, opposite to the pole pieces of the electron and ion columns. It was installed with an angle of 18 degrees to the electron beam, nearly perpendicular to the grid plane. The position of the objective lens can be finely adjusted in three dimensions using an *xyz* piezo stage (Micronix) during focusing.

Cryo-STAR is equipped with an epifluorescence system with a full-spectrum (DAPI–GFP–RFP–Cy5) white LED light source (FC904s, Shanghai Fluoca Technology). The beam excited from the white LED light source is expanded to a diameter of 25 mm, is reflected by a dichroic mirror and passes through into the vacuum chamber. Next, the excitation beam is reflected to the objective lens by a reflector. The emission light from the sample goes back along the incident illumination path and passes through the dichroic mirror and the emission filter. The final image is detected by a high-sensitivity CMOS camera (Moment, Teledyne Photometrics). Micro-Manager (version 2.0)[51] was used to control the camera and record fluorescence images. All lenses used in the system were chosen as achromatic doublets. Optical apertures and filters were used to improve the illumination quality.

### Calibration and operation of the cryo-STAR system

The cryo-STAR system requires precalibration by a finder or index grid with obvious features, such as fluorescent beads. For example, a finder grid is transferred into the SEM chamber by a holder, and then an obvious feature on the grid is found and adjusted to the crossover point of the electron and ion beams. After centering this feature in both the electron and ion images, the position of the objective lens is adjusted to make the feature clear and at the center of the image. In this way, the light beam, the electron beam and the ion beam are adjusted to be at the same focal point, normally with a position accuracy better than 1 µm. The calibration parameters can be checked again during the cryo-FIB milling process to ensure precision.

In the monitoring process, to avoid potential warming up of the cryo-vitrified specimen and induction of devitrification, a stroboscopic exposure mode was used with each exposure time of 200 ms and a camera frame rate of 1–0.5 fps. The total illumination power was controlled at ~20 µW with a minimum illumination intensity of 0.05 W cm⁻². Compared to other cryo-FM techniques, such as cryo-iPALM[24], which always require acquisition of raw image sets with 25,000–75,000 frames or more at a minimum laser power of ~1 W and a minimum illumination intensity of 0.1–1 kW cm⁻², the heating effect induced by our cryo-STAR illumination is minimal, which largely avoids the risk of devitrification of cryo-specimens. Indeed, we did not observe the phenomena of warming up of the frozen specimen and bleaching of the fluorescence signal in our studies.

### Cryo-transfer into ELI-TriScope

Before sample loading, the multispecimen cryo-holder (Gatan) with a homemade tip was cooled to liquid nitrogen temperature. Next, the AutoGrid with the sample was mounted into the tip in the cryo-holder workstation. The position of the objective lens of cryo-STAR was reset to ensure that there was sufficient space above for sample loading. Next, the cryo-holder was inserted into the vacuum transfer system with prepump for approximately 60 s. Next, the mini gate valve was manually opened to allow the cryo-holder to be inserted into ELI-TriScope. After waiting 5 min to recover the high vacuum of the chamber and confirming that the tip of the cryo-holder was far from the pole pieces of the electron and ion columns, the cryo-holder was tilted to 30 degrees

right in front of the GIS[35]. The predefined sample position on the grid was coated with a protective organometallic layer for ~5 s. Next, the cryo-holder was tilted back to −20 degrees, allowing the objective lens of the cryo-STAR system to rise to the focus position for subsequent real-time fluorescence-navigated cryo-FIB milling.

### Culture of HeLa cells and vitrification

HeLa cells expressing mCherry fluorescent protein-labeled pericentrin[33] were seeded onto ultraviolet-sterilized OD grids (T11012SS, TIANLD) and cultured in complete DMEM supplemented with 10% FBS and 1% penicillin and streptomycin. After 48 h of culture with 5% $CO_2$ at 37 °C, the grids were subjected to plunge freezing by backside blotting and vitrification using a Leica EM GP (Leica Microsystems). Next, the cryo-vitrified grid was assembled into an FEI AutoGrid (Thermo Fisher Scientific) and further loaded onto a commercial cryo-EM multiholder (Gatan), ready for the subsequent cryo-transfer.

### Cryo-ET data collection

All cryo-ET data were acquired on an FEI Titan Krios cryo-TEM (Thermo Fisher Scientific) equipped with an energy filter and a K2 Summit Direct electron detector (Gatan) using the SerialEM package[52] at 300 kV. Microscope operation and filter tuning adjustment were performed using SerialEM and Digital Micrograph (Gatan). The slit width of the filter was set to 40 eV. Tilt series were acquired with a unidirectional tilt scheme ranging from −45 to 45 degrees with a step of 2 degrees, and the target defocus was set to −6 µm. The nominal magnification was ×33,000, resulting in a pixel size of 4.3 Å. Individual tilt images were acquired as 3,838 × 3,710-pixel movies including 10–12 frames. All procedures for cryo-EM imaging were performed under low-dose conditions.

### Data processing

Image movies were processed by motion correction and CTF estimation in Warp[53]. The produced tilt series were aligned using IMOD[54] and then transferred back into Warp to reconstruct full tomograms. The filament model in Dynamo[55] was used to manually pick 55 centrioles in 46 tilt series from the proximal end to the distal end, which was judged by the clockwise or counterclockwise direction of MTTs in the cross-sectional view of the centriole[56]. Every particle in the same centriole was separated by 4 nm. All the subvolumes were cropped in Warp, and subsequent data processing, including 3D classification, autorefinement and duplicate removal, was performed in RELION[57]. Different components of the centriole were aligned and refined by shifting the volume center to different local regions (Extended Data Fig. 4). For RD1, RD2, RD3 and DA structures, all the particles were manually picked, and ninefold symmetry was applied during the alignment. To achieve unbiased reconstructions, the templates used in the alignment procedures were all data-driven references, and none of the other reported structures were used in this study.

### Segmentation and visualization

Imaris 9.8.0 (Oxford Instruments) was applied to segment centrioles from two tomograms using a SIRT-like filter equivalent to 15 iterations. The typical characteristics of the centriole were labeled manually and colored differently. All figures were created using ChimeraX[58], and all movies were generated in Imaris 9.8.0.

### Reporting summary

Further information on research design is available in the Nature Portfolio Reporting Summary linked to this article.

### Data availability

The raw tilt series used in this study has been deposited in the Electron Microscopy Public Image Archive China (http://www.emdb-china.org.cn) under accession code EMPIARC-200003 (http://www.emdb-china.org.cn/dataEmpiarc?code=EMPIARC-200003). The cryo-EM maps of

MTTs with a complete C-tubule, MTTs with an incomplete C-tubule, A-tubule, A–C linker and pinhead have been deposited in the Electron Microscopy Database under the accession codes EMD-33417, EMD-33418, EMD-33419, EMD-33420 and EMD-33421, respectively. Source data are provided with this paper.

## Code availability

The LabVIEW program for device controlling is hardware dependent, and the code to control the stage of our ELI-TriScope is available at GitHub: https://github.com/hilbertsun/ELI-TriScope.

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

## Acknowledgements

We give our special thanks to W. Xu for his role in mentoring the team in the field of cryo-EM. We thank F. Wang and J. Chen from Peking University for their kind help in providing HeLa cells, which express mCherry fluorescent protein-labeled pericentrin. We are grateful to X. Huang, B. Zhu, X. Li, L. Sun, L. Qin and Y. Chen for their help with cryo-ET data collection; C. Qi, Y. Teng, Y. Feng, Q. Bian and C. Liu for their help with visualization and segmentation; P. Shan, Y. Jia and S. Li for their assistance with project management. All cryo-FM, cryo-FIB and cryo-ET work was performed in the Center for Biological Imaging (http://www.ibp.cas.cn/cbi/), Institute of Biophysics, Chinese Academy of Science (CAS). This work was equally supported by grants from the National Natural Science Foundation of China (31830020 to F.S.), the Ministry of Science and Technology of China (2021YFA1301501 to F.S. and 2017YFA0504700 to G.J.) and the CAS (XDB37040102 to F.S.). This work was also supported by the Technological Innovation Program of the CAS (29Y8CZ021001 to G.J. and 29Y7CZ041001 to J.Z.) and the CAS Key Technical Support Personnel Project (29Y9CQ041 to G.J.) and by grants from the National Natural Science Foundation of China (31801199 to S.L. and 31801201 to X.J.).

## Author contributions

F.S., G.J. and Y.Z. initiated and supervised the project. G.J. and S.L. designed and built the cryo-STAR system. G.J., J.Z. and S.L. designed and built the cryo-holder-based vacuum transfer system for SEM. G.J. wrote the control software. X.J. and X.Z. performed cell culturing and vitrification. S.L. and G.J. performed cryo-fabrications using the ELI-TriScope system. S.L., X.J. and X.Z. collected cryo-ET tilt series. Y.Z., Z.W., T.N. and G.Y. performed image processing. S.L., Z.W., Y.Z., G.J. and F.S. analyzed the data and wrote the manuscript.

## Competing interests

Part of this study has been assigned a Chinese patent for invention with the numbers CN202110919016.2 and CN202110920369.4.

## Additional information

**Extended data** is available for this paper at https://doi.org/10.1038/s41592-022-01748-0.

**Correspondence and requests for materials** should be addressed to Yun Zhu, Gang Ji or Fei Sun.

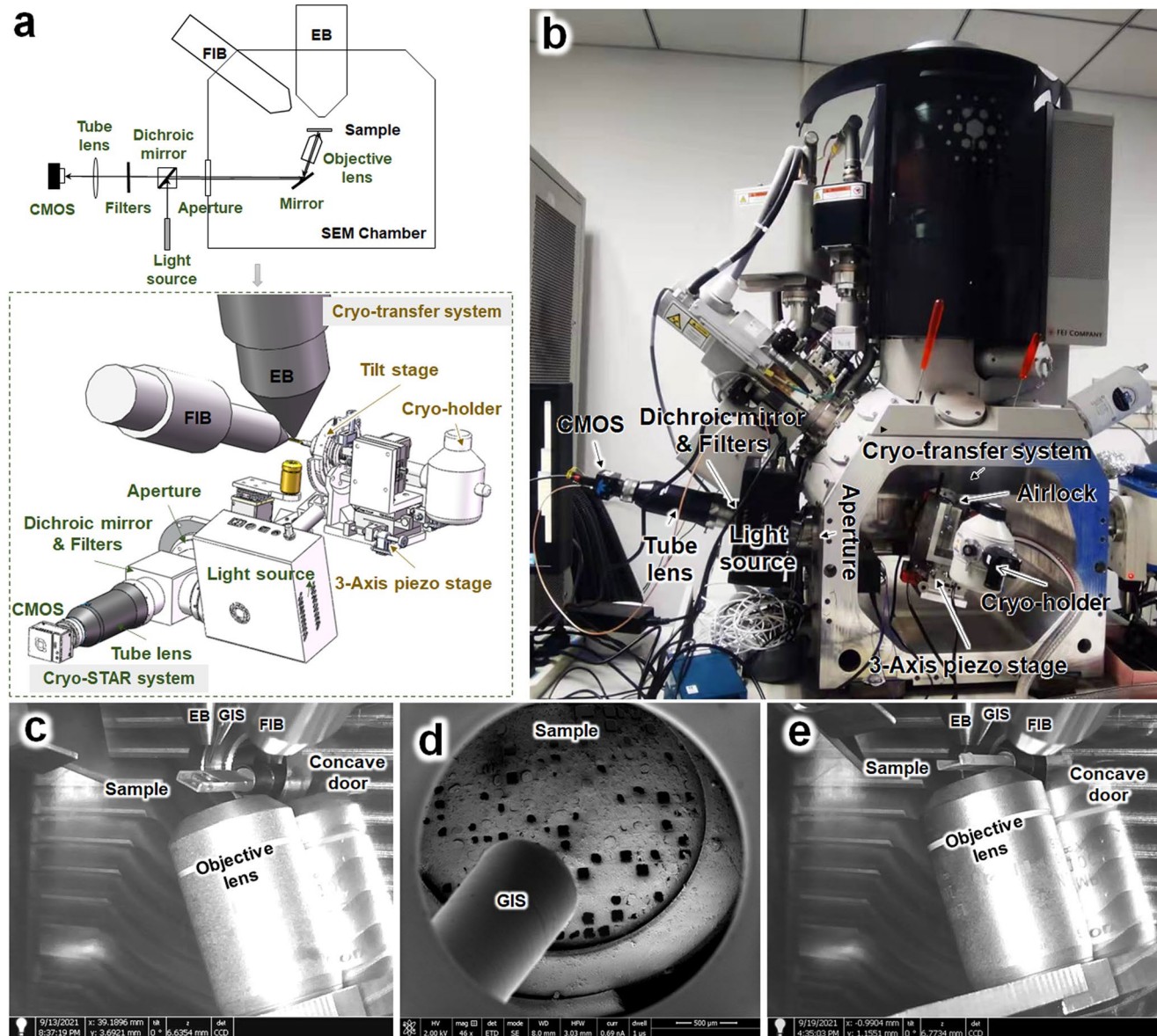

**Extended Data Fig. 1 | Design and construction of ELI-TriScope. (a)** Schematic diagram of the ELI-TriScope system showing two major components, a cryo-holder-based cryo-transfer system and a cryo-STAR system. Along the direction of the optical path of the cryo-STAR system, the excitation light beam coming out of the light source is reflected by a dichroic mirror into the aperture, reflected into the objective lens by an adjustable mirror and then excites the fluorescent molecules in the cryo-specimen. Next, the excited fluorescence is received by the objective lens, is reflected into the aperture by the adjustable mirror, passes through a dichroic mirror, is filtered and is focused on a CMOS camera through a tube lens. The cryo-transfer system is equipped with a 3D motorized device that is adapted to the cryo-holder. **(b)** Photograph of our built ELI-TriScope system based on FEI Helios NanoLab 600i. A custom-designed cryo-transfer system replaces the original chamber door, and the cryo-STAR system is fixed on the other side of the chamber. **(c)** Photograph of the internal architecture of ELI-TriScope. The cryo-holder is tilted 30 degrees to the right to face the GIS for coating. **(d)** Observation of the coating status by SEM imaging at 40X magnification after GIS coating. **(e)** Position for cryo-FIB fabrication. The cryo-holder is tilted to −15 degrees. The objective lens of cryo-STAR rises to the focus position. Each part of the system is indicated and labeled. EB, electron beam. GIS, gas injection system.

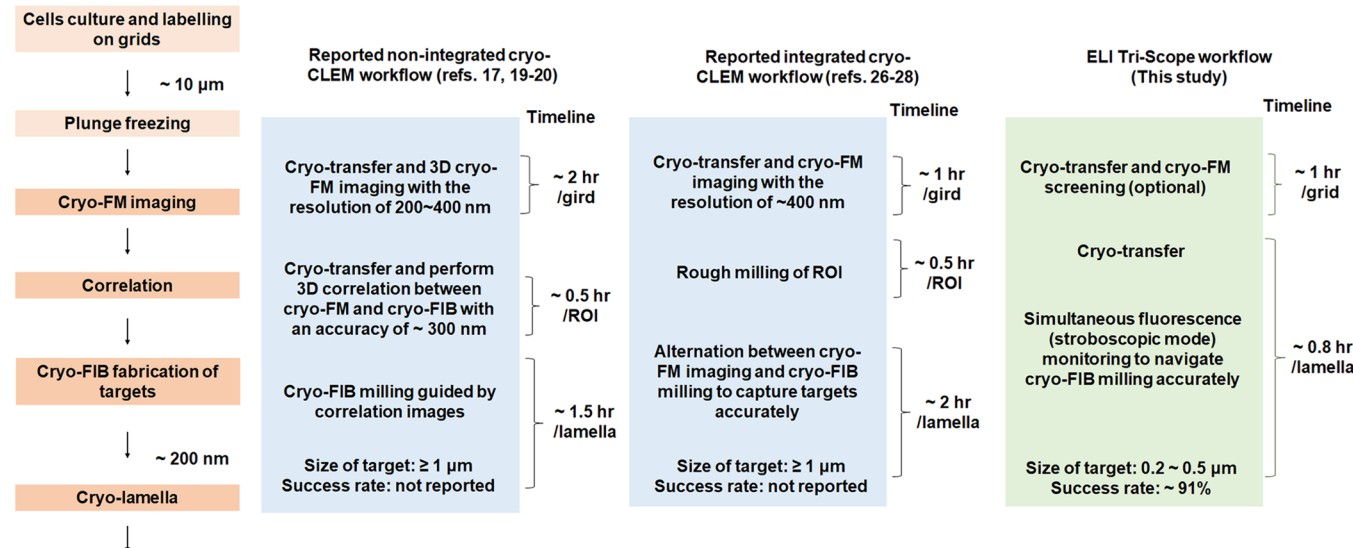

**Extended Data Fig. 2 | Comparison of three different cryo-CLEM workflows for site-specific cryo-FIB fabrication.** A conventional experimental procedure of the in situ cryo-ET study is described (left), including steps from cell culture to plunge freezing, cryo-FM imaging, cryo-FIB imaging and correlation, cryo-FIB milling, production of a cryo-lamella and cryo-ET data collection. The reported cryo-CLEM workflows from cryo-FM imaging to cryo-FIB milling can be classified into nonintegrated workflows[17,19,20] and integrated workflows[26–28]. In a nonintegrated workflow, the cryo-FM and cryo-FIB steps are carried out individually and sequentially. Then, with the help of fiducial markers, cryo-FM images and cryo-FIB images are aligned by 3D correlation software and used to guide cryo-FIB milling. In an integrated workflow, such as in the commercial products iFLM (FEI) and METEOR (Delmic), due to the limited space inside the chamber and the limitation set by the cryo-stage, only a low-NA objective lens can be installed just alongside the ion column. Cryo-FM images are alternatively acquired before and after cryo-FIB milling to check the target fluorescence signal and accurately guide cryo-FIB milling, which reduces the overall throughput of cryo-lamella production. For both nonintegrated and integrated workflows, with the cryo-FM resolution and correlation accuracy limitations, the sizes of the studied targets (for example, mitochondria, lipid droplets) are normally larger than 1 μm, and the success rate of cryo-lamella production was not reported. In the ELI-TriScope workflow, after the optional cryo-FM imaging, there is no need for a 3D correlation step, and the fluorescence signal can be acquired in real-time stroboscopic mode to simultaneously monitor the cryo-FIB milling procedure. With this, the time cost to prepare one cryo-lamella using the ELI-TriScope workflow can be greatly reduced to ~0.8 hrs per lamella, in comparison with 2–2.5 hrs per cryo-lamella for previously reported nonintegrated and integrated workflows.

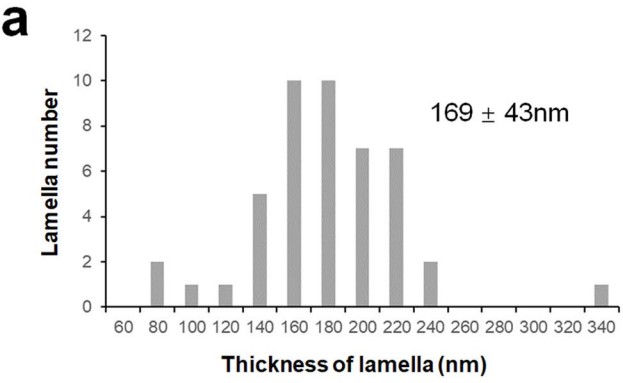

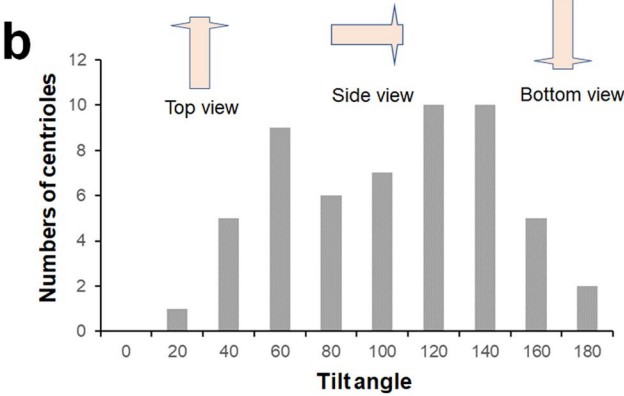

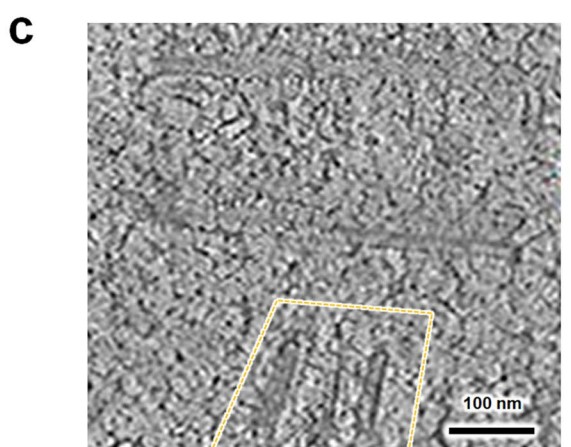

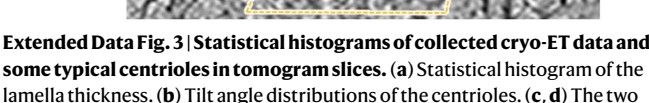

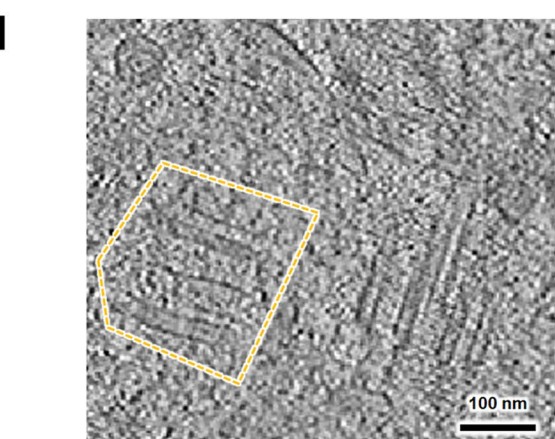

**Extended Data Fig. 3 | Statistical histograms of collected cryo-ET data and some typical centrioles in tomogram slices.** (**a**) Statistical histogram of the lamella thickness. (**b**) Tilt angle distributions of the centrioles. (**c**, **d**) The two newborn human procentrioles observed in the dataset. Slice views of possible newborn procentrioles (sandy brown arrows) were observed in different tomograms.

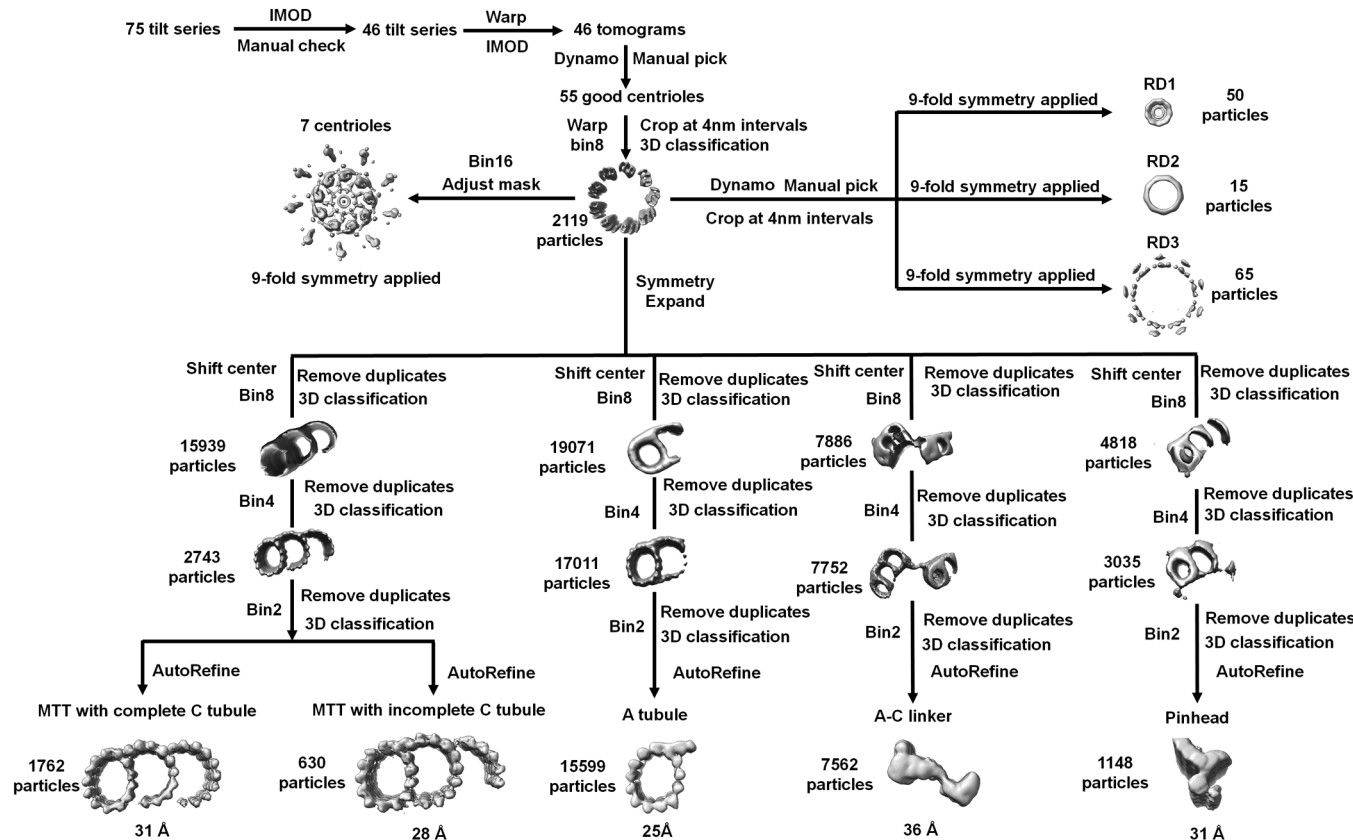

**Extended Data Fig. 4 | Cryo-ET data processing workflow for different parts of the human centriole.**

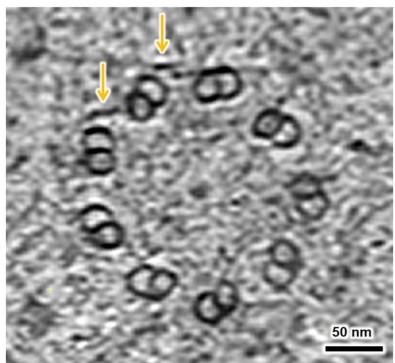

**Extended Data Fig. 5 | MTD structure of the human centriole in a slice view of a tomogram.** The missing or incomplete C-tubules are indicated by yellow arrows. Only one centriole with representative MTD structure was observed in the dataset.

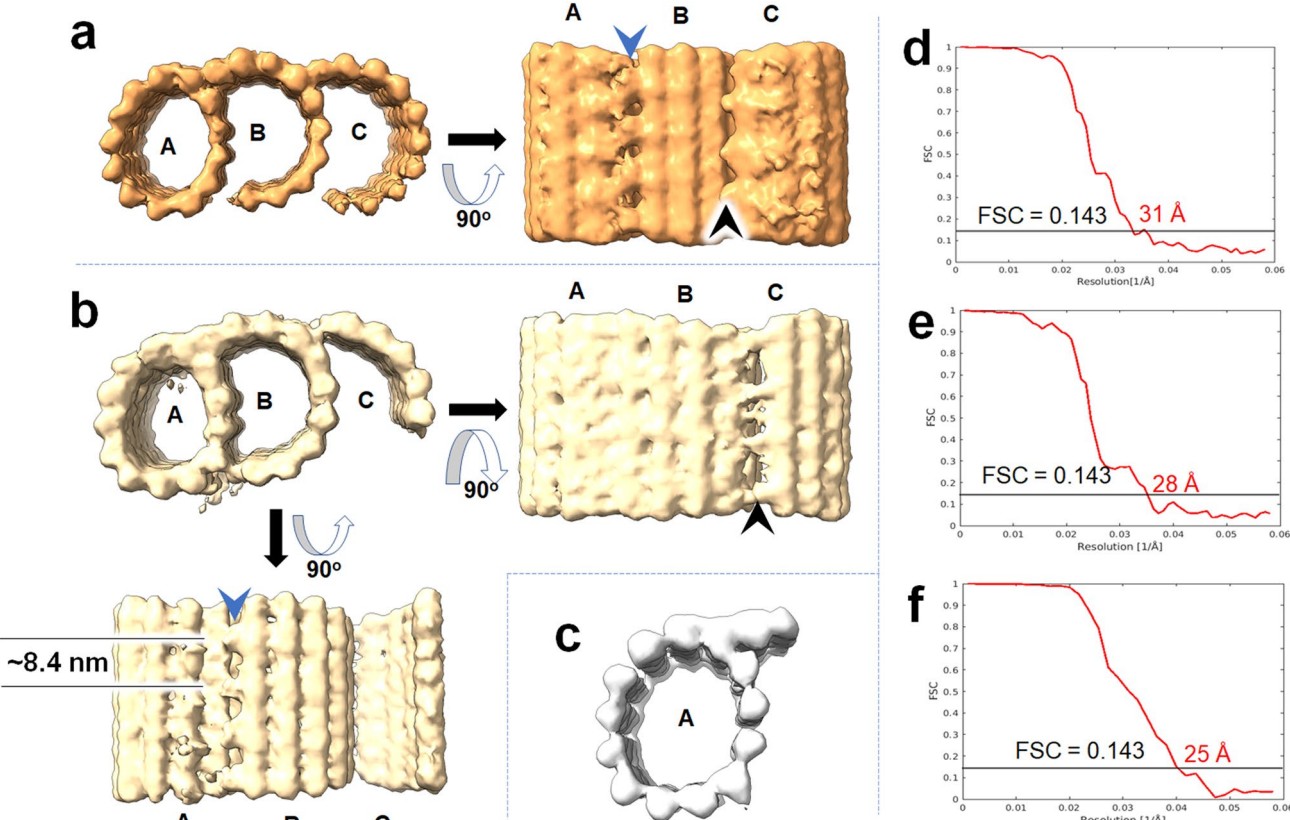

**Extended Data Fig. 6 | STA of MTTs from the human centriole. (a)** Top view and side view of MTTs with a complete C-tubule, showing junctions between the A- and B-tubules (blue arrow) and B- and C-tubules (black arrow). **(b)** Top view and two side views of MTTs with an incomplete C-tubule, showing junctions between the A- and B-tubules (blue arrow) and B- and C-tubules (black arrow). **(c)** Top view of a single A-tubule. The gold-standard Fourier shell correlation (FSC) curves for the MTTs in **(a)**, **(b)** and **(c)** are shown in **(d)**, **(e)** and **(f)**, respectively. The resolutions are calculated according to the FSC 0.143 criterion and labeled.

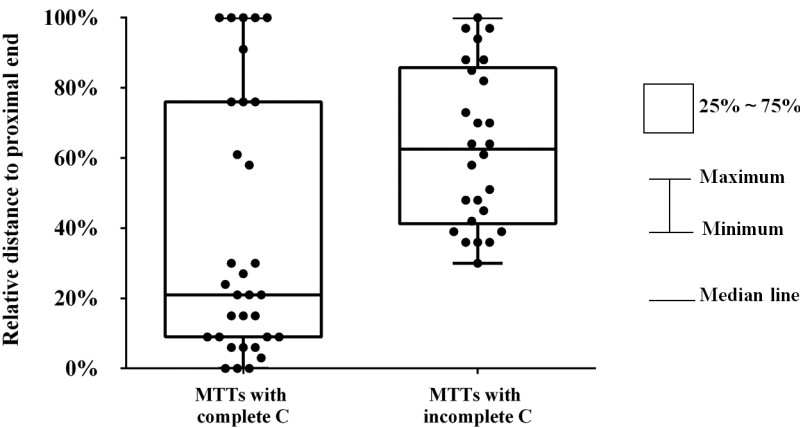

**Extended Data Fig. 7 | Distributions of MTTs with a complete C-tubule and an incomplete C-tubule in a single centriole.** The two ends of the centriole in the tomogram are taken for the relative proximal and distal ends. The coordinates of the refined MTT elements are projected onto the central symmetry axis of the centriole to calculate the relative distances (0–100%). The minimal and maximal relative distances in each statistic are indicated as edge bars, respectively. The box boundaries represent interquartile ranges (25–75%). The line in the middle of the box represents the median distance. Since most centrioles in our dataset are incomplete ones, a representative long centriole was chosen (n = 1) to calculate distributions of MTTs.

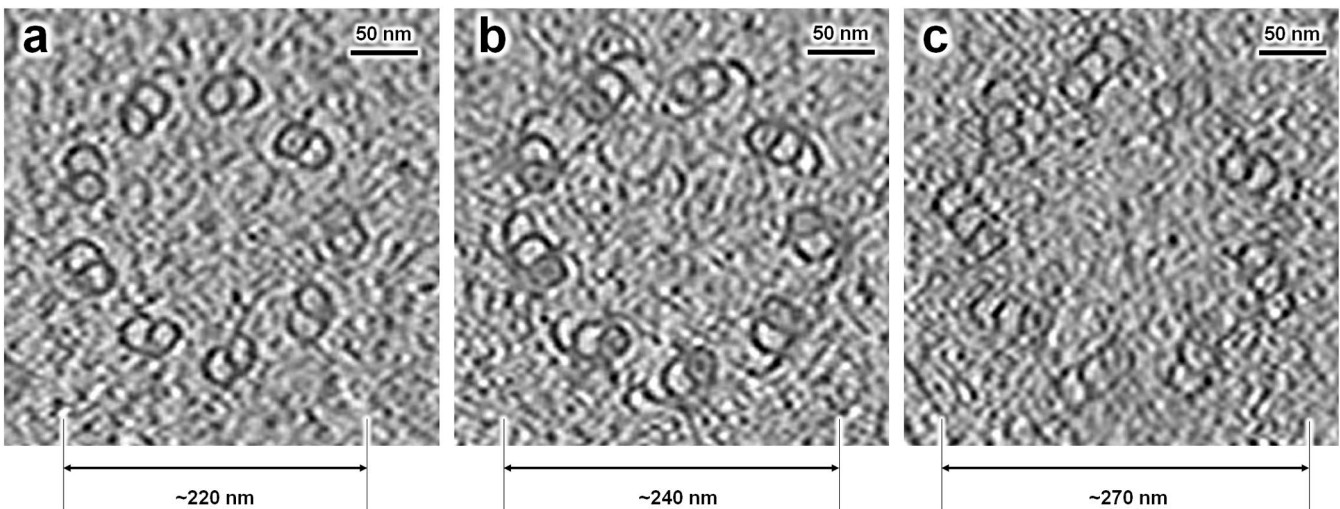

**Extended Data Fig. 8 | Different diameters of human centrioles in the slice view of tomograms.** (**a**) The tomogram slice shows one representative centriole with ~220 nm diameter selected from two observed ones. (**b**) The tomogram slice shows one representative centriole with ~240 nm diameter selected from five observed ones. (**c**) The tomogram slice shows one representative centriole with ~270 nm diameter selected from twelve observed ones.

**Extended Data Table 1 | Statistics of cryo-ET data collection and image processing**

| Data acquisition | |
|---|---|
| Microscope | Titan Krios G2 |
| Voltage (kV) | 300 |
| Detector | Gatan K2 |
| Energy filter | Gatan GIF Quantum, 40 eV |
| Mode | Counting |
| Pixel size (Å) | 4.3 |
| Stage tilting angle | -45° - 45° |
| Number of images | 45 - 60 |
| Exposure per image(e/Å²) | 3 |
| Exposure per tilt(e/Å²) | 120 - 140 |
| Defocus range (μm) | -6 - -10 |
| Software | SerialEM |

| Sub tomogram analysis | | | | | |
|---|---|---|---|---|---|
| Software | RELION-3.1/Warp | | | | |
| Data set | MTTs with complete C | MTTs with incomplete C | A-tubule | A-C linker | pinhead |
| Number of particles | 1762 | 630 | 15599 | 7562 | 1148 |
| Symmetry | C1 | C1 | C1 | C1 | C1 |
| Resolution (Å) | 31 | 28 | 25 | 35.9 | 31 |
| Map pixel size (Å) | 8.6 | 8.6 | 8.6 | 8.6 | 8.6 |

# nature research

# Reporting Summary

Nature Research wishes to improve the reproducibility of the work that we publish. This form provides structure for consistency and transparency in reporting. For further information on Nature Research policies, see our Editorial Policies and the Editorial Policy Checklist.

## Statistics

For all statistical analyses, confirm that the following items are present in the figure legend, table legend, main text, or Methods section.

| n/a | Confirmed | |
|---|---|---|
| ☐ | ☒ | The exact sample size (*n*) for each experimental group/condition, given as a discrete number and unit of measurement |
| ☐ | ☒ | A statement on whether measurements were taken from distinct samples or whether the same sample was measured repeatedly |
| ☒ | ☐ | The statistical test(s) used AND whether they are one- or two-sided<br>*Only common tests should be described solely by name; describe more complex techniques in the Methods section.* |
| ☒ | ☐ | A description of all covariates tested |
| ☒ | ☐ | A description of any assumptions or corrections, such as tests of normality and adjustment for multiple comparisons |
| ☐ | ☒ | A full description of the statistical parameters including central tendency (e.g. means) or other basic estimates (e.g. regression coefficient) AND variation (e.g. standard deviation) or associated estimates of uncertainty (e.g. confidence intervals) |
| ☒ | ☐ | For null hypothesis testing, the test statistic (e.g. *F*, *t*, *r*) with confidence intervals, effect sizes, degrees of freedom and *P* value noted<br>*Give P values as exact values whenever suitable.* |
| ☒ | ☐ | For Bayesian analysis, information on the choice of priors and Markov chain Monte Carlo settings |
| ☒ | ☐ | For hierarchical and complex designs, identification of the appropriate level for tests and full reporting of outcomes |
| ☒ | ☐ | Estimates of effect sizes (e.g. Cohen's *d*, Pearson's *r*), indicating how they were calculated |

*Our web collection on statistics for biologists contains articles on many of the points above.*

## Software and code

Policy information about availability of computer code

| Data collection | Cryo-ET data were automatically collected using SerialEM3.8 at Center for Biological Imaging (CBI, http://cbi.ibp.ac.cn), Institute of Biophysics, Chinese Academy of Sciences. Micro-Manager (ver 2.0) was used to control the camera and record fluorescence images. The movement and tilt of the stage can be controlled via customized software written in LabVIEW 2011 (National Instruments, USA). The LabVIEW program for device controlling is hardware-dependent and the code to control the stage of our ELI-TriScope is available at the GitHub: https://github.com/hilbertsun/ELI-TriScope. |
|---|---|
| Data analysis | Cryo-EM data were analyzed using the RELION3.1, RELION3.0, UCSF-ChimeraX 1.3, IMOD 4.12.16, Imaris 9.8.0, Warp1.0.9. |

For manuscripts utilizing custom algorithms or software that are central to the research but not yet described in published literature, software must be made available to editors and reviewers. We strongly encourage code deposition in a community repository (e.g. GitHub). See the Nature Research guidelines for submitting code & software for further information.

## Data

Policy information about availability of data

All manuscripts must include a data availability statement. This statement should provide the following information, where applicable:

- Accession codes, unique identifiers, or web links for publicly available datasets
- A list of figures that have associated raw data
- A description of any restrictions on data availability

The raw tilt series used in this study has been deposited in EMPIAR (the Electron Microscopy Public Image Archive) China (http://www.emdb-china.org.cn) under accession code EMPIARC-200003. The sub-tomogram averaged cryo-EM maps of the MTTs with complete C tubule, MTTs with incomplete C tubule, A tubule, A-C linker and pinhead have been deposited in the Electron Microscopy Database (EMDB) with the accession codes EMD-33417, EMD-33418, EMD-33419, EMD-33420 and EMD-33421, respectively. All other data that support the conclusion of this study are provided in the supplementary data and source data.

# Field-specific reporting

Please select the one below that is the best fit for your research. If you are not sure, read the appropriate sections before making your selection.

☒ Life sciences  ☐ Behavioural & social sciences  ☐ Ecological, evolutionary & environmental sciences

For a reference copy of the document with all sections, see nature.com/documents/nr-reporting-summary-flat.pdf

# Life sciences study design

All studies must disclose on these points even when the disclosure is negative.

| | |
|---|---|
| Sample size | We prepared 40 vitrified cryo-grids, and 21 of them showed good quality (proper ice thickness and well distributed fluorescent signals) in the cryo-FM screening and were used for ELI-TriScope milling. We fabricated a total of 79 cryo-lamellae, and found 72 of them containing centrioles, which were used for the subsequent cryo-ET data collection. During sub-tomogram averaging process, the number of final particles that went into each refined map were determined through 3D classification as described in Extended Data Fig. 4. The sample size is limited by the resources of microscope time and the current size has been enough to determine the successful rate of our ELI-TriScope workflow. |
| Data exclusions | Tomograms with poor quality or unable to be aligned were excluded. |
| Replication | To measure the successful rate of our ELI-TriScope workflow, we repeated performing ELI-TriScope milling of 21 vitrified grids and the number of repeating ELI-TriScope workflow is 79 and we found 72 of them were successful to target the regions containing centrioles. |
| Randomization | Randomization was not applicable because there was nothing related to comparison between control and experimental groups in this study. |
| Blinding | Blinding was not applicable because there was nothing related to comparison among different groups in this study. |

# Reporting for specific materials, systems and methods

We require information from authors about some types of materials, experimental systems and methods used in many studies. Here, indicate whether each material, system or method listed is relevant to your study. If you are not sure if a list item applies to your research, read the appropriate section before selecting a response.

## Materials & experimental systems

| n/a | Involved in the study |
|---|---|
| ☒ | ☐ Antibodies |
| ☐ | ☒ Eukaryotic cell lines |
| ☒ | ☐ Palaeontology and archaeology |
| ☒ | ☐ Animals and other organisms |
| ☒ | ☐ Human research participants |
| ☒ | ☐ Clinical data |
| ☒ | ☐ Dual use research of concern |

## Methods

| n/a | Involved in the study |
|---|---|
| ☒ | ☐ ChIP-seq |
| ☒ | ☐ Flow cytometry |
| ☒ | ☐ MRI-based neuroimaging |

# Eukaryotic cell lines

Policy information about cell lines

| | |
|---|---|
| Cell line source(s) | HeLa (ATCC: CCL-2) cells |
| Authentication | HeLa (ATCC: CCL-2) cells were obtained from ATCC and provided from Prof. Jianguo Chen's lab in Peking University and the cell lines were made according to the literature (EMBO Reports 1, 524-529) |
| Mycoplasma contamination | Not detected |
| Commonly misidentified lines (See ICLAC register) | No commonly misidentified cell lines were used. |

