## [Peer Review File · Nature Methods]

Peer Review Information

Manuscript Title: ELI trifocal microscope: A precise cryogenic fabrication system to prepare target cryo-lamellae of cells for in situ cryo-ET study

Corresponding author name(s): Yun Zhu, Gang Ji, Fei Sun

Editorial Notes:

Reviewer Comments & Decisions:

Decision Letter, initial version:

Dear Fei,

Please let me begin by apologizing for the time your paper spent in peer review. To prevent the time from getting longer, we have made our decision based on two reviews.

Your Article, "ELI trifocal microscope: A precise cryogenic fabrication system to prepare target cryo-lamellae of cells for in situ cryo-ET study", has now been seen by two reviewers. As you will see from their comments below, although the reviewers find your work of considerable potential interest, they have raised a number of concerns. We are interested in the possibility of publishing your paper in Nature Methods, but would like to consider your response to these concerns before we reach a final decision on publication.

We therefore invite you to revise your manuscript to address these concerns. This includes addressing technical concerns, making some textual changes, and sharing as much representative data as possible.

* include a point-by-point response to the reviewers and to any editorial suggestions

* please underline/highlight any additions to the text or areas with other significant changes to facilitate review of the revised manuscript

* address the points listed described below to conform to our open science requirements

* ensure it complies with our general format requirements as set out in our guide to authors at www.nature.com/naturemethods

* resubmit all the necessary files electronically by using the link below to access your home page

[Redacted] This URL links to your confidential home page and associated information about manuscripts you may have submitted, or that you are reviewing for us. If you wish to forward this email to co-authors, please delete the link to your homepage.

We hope to receive your revised paper within two months. If you cannot send it within this time, please let us know. In this event, we will still be happy to reconsider your paper at a later date so long as nothing similar has been accepted for publication at Nature Methods or published elsewhere.

OPEN SCIENCE REQUIREMENTS

REPORTING SUMMARY AND EDITORIAL POLICY CHECKLISTS

Please note that these forms are dynamic ‘smart pdfs’ and must therefore be downloaded and completed in Adobe Reader. We will then flatten them for ease of use by the reviewers. If you would like to reference the guidance text as you complete the template, please access these flattened versions at <http://www.nature.com/authors/policies/availability.html>.

DATA AVAILABILITY

All novel DNA and RNA sequencing data, protein sequences, genetic polymorphisms, linked genotype and phenotype data, gene expression data, macromolecular structures, and proteomics data must be deposited in a publicly accessible database, and accession codes and associated hyperlinks must be provided in the “Data Availability” section.

Please include a “Data availability” subsection in the Online Methods. This section should inform readers about the availability of the data used to support the conclusions of your study, including accession codes to public repositories, references to source data that may be published alongside the paper, unique identifiers such as URLs to data repository entries, or data set DOIs, and any other statement about data availability. At a minimum, you should include the following statement: “The data that support the findings of this study are available from the corresponding author upon request”, describing which data is available upon request and mentioning any restrictions on availability. If DOIs are provided, please include these in the Reference list (authors, title, publisher (repository name),

identifier, year). For more guidance on how to write this section please see:

<http://www.nature.com/authors/policies/data/data-availability-statements-data-citations.pdf>

CODE AVAILABILITY

Please include a “Code Availability” subsection in the Online Methods which details how your custom code is made available. Only in rare cases (where code is not central to the main conclusions of the paper) is the statement “available upon request” allowed (and reasons should be specified).

MATERIALS AVAILABILITY

ORCID

Nature Methods is committed to improving transparency in authorship. As part of our efforts in this direction, we are now requesting that all authors identified as ‘corresponding author’ on published papers create and link their Open Researcher and Contributor Identifier (ORCID) with their account on the Manuscript Tracking System (MTS), prior to acceptance. This applies to primary research papers only. ORCID helps the scientific community achieve unambiguous attribution of all scholarly contributions. You can create and link your ORCID from the home page of the MTS by clicking on ‘Modify my Springer Nature account’. For more information please visit www.springernature.com/orcid.

Sincerely,
Rita

Rita Strack, Ph.D.
Senior Editor
Nature Methods

Reviewers' Comments:

Reviewer #1:
None

Reviewer #2:

Remarks to the Author:

In this manuscript, Li et al, describe a novel tri-focal microscope in which they have integrated a light objective for fluorescence excitation and detection within the chamber of a dual-beam microscope. This paper was easy to read, and the process was well explained. The real novelty comes in the fact that this objective lens is calibrated to be in focal alignment with both the SEM and FIB beams, which allows for real time monitoring of fluorescence signal during the production of vitreous lamella for tomography. The authors chose a suitably difficult object to target (the centrosome), as only one pair exists per non-dividing cell, and they are only a few hundred nanometers in all dimensions. After milling and collecting data from 60+ lamella, they achieved a very high success rate. I think this study would be of great interest to the cryo-EM community, and suggest you publish it in Nature Methods.

Minor Concern:

In the first paragraph of the discussion the authors claim:

“Our cryo-transfer system achieves seamless and contactless cryo-specimen transfer among our HOPE cryo-FM system, ELI-TriScope system and FEI Titan Krios cryo electron microscope and largely decreases the risk of specimen damage, deformation, devitrification, and ice contamination.”

From what I can tell, the transfer is neither seamless nor contactless. Potentially, there is a reduced number for transfers, due to the integrated microscope, but this is not novel. Additionally, you must still transfer out of the ELI scope to the Krios, and given they are using a side-entry holder it will be essential to remove the clipped grid from the holder and put it into the Krios cassette. The authors should be more explicit about how they have altered their protocol to be “seamless and contactless” or leave such a statement out.

Reviewer #3:

Remarks to the Author:

This paper has creatively developed a new platform by incorporating an additional light beam to the traditional SEM/FIB system with their new design of high-vacuum optical fluorescence microscope. The three-beam system of electron, light, and ion enables targeting of region of interest more accurately and efficiently compared to both current commercial devices of iFLM and METEOR. I am very impressed by the time for producing one lamella shortened from 2~2.5 to 0.8 hrs, although not very convinced by their statements of no ice contamination, no or less specimen damage, and 100% success ratio since no data has been presented in the paper. The authors have also performed experiment on human centriole and presented detailed and convincing tomogram results. These results are partially consistent with previously published results, or even newer. I am hoping to see its application on higher resolution in sub-nano range. In summary, the paper was well written based on their detailed experiment and results. The method they developed is very interesting and becomes a breakthrough on the current existing Cryo-FIB devices.

Although there are few points below which need to be further clarified:

- (1) The authors claimed 100% success ratio of FIB milling, which is quite unbelievable. Please present all possible experimental data if it is true.
- (2) The authors claimed no ice contamination for any lamella they made, please at least present all bright-field images of those 61 lamellas to prove this statement.
- (3) In Figure 3b and 3c, both sub-tomogram averaged maps of RD1 and RD2 show very different structures from those in raw tomograms. Authors would be very careful when they applied 9-fold symmetry and not overstate the averaged map especially with low signal-to-noise ratio data.
- (4) The fiducial beads free is the one advantage that the authors claimed in the paper, while fiducial beads are good reference especially when correlating the fluorescent image to cryoEM and help localize the region of interesting. Besides I haven't seen any statement of how the ROI are localized in cryoEM, if

it is manual then it will be a big limitation for the application of this method to achieve high resolution, especially for small particles.

Author Rebuttal to Initial comments

Response to Reviewers

Reviewer #1:

None.

Reviewer #2:

Remarks to the Author:

Comment 1: *In this manuscript, Li et al, describe a novel tri-focal microscope in which they have integrated a light objective for fluorescence excitation and detection within the chamber of a dual-beam microscope. This paper was easy to read, and the process was well explained. The real novelty comes in the fact that this objective lens is calibrated to be in focal alignment with both the SEM and FIB beams, which allows for real time monitoring of fluorescence signal during the production of vitreous lamella for tomography. The authors chose a suitably difficult object to target (the centrosome), as only one pair exists per non-dividing cell, and they are only a few hundred nanometers in all dimensions. After milling and collecting data from 60+ lamella, they achieved a very high success rate. I think this study would be of great interest to the cryo-EM community, and suggest you publish it in Nature Methods.*

Response 1:

We would like to thank this reviewer for his/her professional comment and high evaluation of our work. We believe our work will be helpful for the *in situ* structural study of the cryo-EM community.

Minor Concern:

Comment 2: *In the first paragraph of the discussion the authors claim:*

“Our cryo-transfer system achieves seamless and contactless cryo-specimen transfer among our HOPE cryo-FM system, ELI-TriScope system and FEI Titan Krios cryo electron microscope and

largely decreases the risk of specimen damage, deformation, devitrification, and ice contamination.”

From what I can tell, the transfer is neither seamless nor contactless. Potentially, there is a reduced number for transfers, due to the integrated microscope, but this is not novel. Additionally, you must still transfer out of the ELI scope to the Krios, and given they are using a side-entry holder it will be essential to remove the clipped grid from the holder and put it into the Krios cassette. The authors should be more explicit about how they have altered their protocol to be “seamless and contactless” or leave such a statement out.

Response 2:

We apologize for the ambiguity in the statements. We intended to tell during the procedure from our HOPE cryo-FM system, ELI-TriScope system and FEI Titan Krios cryo-electron microscope, the EM grid with the specimen on is never touched since it is loaded into FEI AutoGrid and protected at the beginning of the workflow. This is the meaning of “contactless”. We agree that our cryo-transfer system is not completely seamless because we still need to transfer the AutoGrid from the side-entry cryo-holder to the Krios cassette. As this reviewer mentioned, our cryo-CLEM workflow can reduce the number of specimen transfers. During the screening process in the HOPE cryo-FM system and the milling process in the ELI-TriScope system, the cryo-specimen is kept in the same cryo-holder all the time. To avoid any potential ambiguity as this reviewer suggested, we have revised the statements in the manuscript as follows:

“Our cryo-transfer system achieves seamless and contactless cryo-specimen transfer between our HOPE cryo-FM system and ELI-TriScope system. The touch of the EM grid with the specimen can be largely avoided during the whole cryo-CLEM workflow. The reduced number of specimen transfers largely decreases the risk of specimen damage, deformation, devitrification, and ice contamination”. **(Page 16, Line 17-22).**

Reviewer #3:

Remarks to the Author:

Comment 3: *This paper has creatively developed a new platform by incorporating an additional light beam to the traditional SEM/FIB system with their new design of high-vacuum optical*

fluorescence microscope. The three-beam system of electron, light, and ion enables targeting of region of interest more accurately and efficiently compared to both current commercial devices of iFLM and METEOR. I am very impressed by the time for producing one lamella shortened from 2~2.5 to 0.8 hrs, although not very convinced by their statements of no ice contamination, no or less specimen damage, and 100% success ratio since no data has been presented in the paper.

The authors have also performed experiment on human centriole and presented detailed and convincing tomogram results. These results are partially consistent with previously published results, or even newer. I am hoping to see its application on higher resolution in sub-nano range. In summary, the paper was well written based on their detailed experiment and results. The method they developed is very interesting and becomes a breakthrough on the current existing Cryo-FIB devices.

Response 3:

We would like to thank this reviewer for his/her high appraisal and enthusiastic point of our work, and for his/her fruitful suggestions and comments, which have helped us to improve the quality of our manuscript. We believe our work will become useful for the *in situ* structural study of the cryo-EM community. For the statements of ice contamination, specimen damage and success ratio, we give our detailed responses in the following (see Responses #4 and #5). In consistency with the dream of this reviewer, our primary motivation of developing ELI-TriScope is to solve the high resolution *in situ* structure of human centriole. With the success of ELI-TriScope, the efficiency and throughput of cryo-specimen preparation have been largely increased. And now we are ready to work hard to push the resolution into sub-nano range and to see secondary structure, which is just undergoing. We need to collect more folds of datasets to achieve this goal and will publish our results in the future.

Although there are few points below which need to be further clarified:

Comment 4: *(1) The authors claimed 100% success ratio of FIB milling, which is quite unbelievable. Please present all possible experimental data if it is true.*

Response 4:

We apologize for not providing experimental data statistics and therefore making a potential confusion in the original manuscript. In our current study (see Fig.2e in the revision), we prepared 40 vitrified cryo-grids, and 21 of them showed good quality (proper ice thickness and well distributed fluorescent signals) in the cryo-FM screening and were used for ELI-TriScope milling. We fabricated a total of 79 cryo-lamellae, and found 72 of them containing centrioles, which were used for the subsequent cryo-ET data collection. As a result, based on our current statistics, we calculated the success rate of targeting centrioles by using ELI-TriScope as ~91% (72/79). This is the reason why we claimed a close to 100% (~100%) success ratio. We did not intend to claim the success ratio of cryo-FIB milling itself but want to emphasize the high success ratio of targeting to ROI by ELI-TriScope. Conventional cryo-CLEM workflow could not achieve such high success ratio of targeting. In our recent study, we found the success ratio of targeting by the conventional cryo-CLEM workflow is significantly attenuated by a kind of effect we called as cryo-FIB milling induced distortion. We are currently performing systematic study of this effect and will publish our results in the future.

In the revised manuscript, we have provided all the related experimental data of cryo-FIB milling (Extended Data Figure 9 and Supplementary Dataset 2). The list of cryo-EM images of all the cryo-lamellae fabricated in this study are summarized in Extended Data Figure 9 with the locations of centrioles indicated and the averaged power spectrums of corresponding tilt series are also given. The corresponding raw cryo-EM micrographs are also provided with the locations of centrioles indicated in Supplementary Dataset 2.

To avoid any potential confusion and make a rigorous statement, we have revised the text as follows,

“...we prepared a batch of cryo-lamellae of HeLa cells targeting the centrosome, with a success rate of ~ 91%...” (**Page 2, Line 16-17**)

“A batch of cryo-lamellae of HeLa cells targeting the centrosome were efficiently prepared with a success rate of ~ 91%.” (**Page 5, Line 21-23**)

“In this study, we prepared 40 vitrified cryo-grids, and 21 of them showed good quality (proper ice thickness and well distributed fluorescent signals) in the cryo-FM screening and were used for ELI-TriScope milling. We fabricated a total of 79 cryo-lamellae, and found 72 of them containing centrioles, which were used for the subsequent cryo-ET data collection (Fig. 2e, Extended Data Fig. 9 and Supplementary Dataset 2). As a result, based on our current statistics,

we calculated the success rate of targeting centrioles by using ELI-TriScope as ~91%.” (Page 10, Line 24-26 & Page 11, Line 1-5)

Comment 5: (2) *The authors claimed no ice contamination for any lamella they made, please at least present all bright-field images of those 61 lamellas to prove this statement.*

Response 5:

We apologize for not providing experimental data statistics and therefore making a potential confusion in the original manuscript. As we response to **Reviewer #2**, our cryo-CLEM workflow can reduce the number of specimen transfers, which largely decreases the risk of specimen damage, deformation, devitrification, and ice contamination. However, it's not practical to achieve “no ice contamination for any lamellae”. In our manuscript, we prefer to use the statements like “the risk of ice contamination...can be largely avoided” and “minimizing the risk of ...ice contamination”.

To support our statements, in the revision, we have provided all cryo-EM images of the cryo-lamellae fabricated in this study (see Extended Data Figure 9 and Supplementary Dataset 2). Our statistics showed that at the regions of cryo-ET data collection ~ 83% of cryo-lamellae show little or no ice contamination, ~13% with slight ice contamination, while the rest ~ 4% contain severe ice contamination (Extended Data Figure 9). We have deposited all the raw tilt series of cryo-ET data in EMPIAR (the Electron Microscopy Public Image Archive) China (<http://www.emdb-china.org.cn>) under accession code EMPIARC-200003, allowing this reviewer and further readers to further evaluate the quality of our dataset.

Comment 6: (3) *In Figure 3b and 3c, both sub-tomogram averaged maps of RD1 and RD2 show very different structures from those in raw tomograms. Authors would be very careful when they applied 9-fold symmetry and not overstate the averaged map especially with low signal-to-noise ratio data.*

Response 6:

We would like to thank this reviewer for his/her professional suggestions and comments and we agree that we should be very careful when we applied 9-fold symmetry and should not overstate the averaged map.

Compared with MTTs, the inner region of centriole presents a variety of different structures, resulting in a lower number of particles in each type. Moreover, the inner region showed little feature for the centrioles in the tilt and side views. Therefore, it's difficult to validate by eyes whether the classified particles of inner region are correctly assigned. In the revision, we manually picked all top-view particles corresponding to RD1, RD2, and RD3 structures, and aligned them separately using both C1 symmetry and C9 symmetry. As shown in Fig. R1, the small number of particles yielded a low resolution and noisy map for C1 symmetry but can reconstruct a better map with the C9 symmetry, which looks in consistency with its original density in the raw tomograms (see revised Figs. 3b-d).

Figure R1. The top view (left) and slice view (right) of averaged reconstructions from manually picked RD1, RD2, and RD3 particles.

By analyzing the original tomograms, we found that many RD structures inside the centrioles are incomplete due to the cryo-FIB milling. Together with the small number of particles, the alignment using C1 symmetry became very difficult. Therefore, we decide to use the averaged maps with C9-symmetry applied to show the possible shape of RD1, RD2, and RD3 structures (see revised Fig. 3 and Extended Data Fig. 4). The following sentences are also revised to avoid misunderstanding and overstatement.

“The unambiguous top-view particles corresponding to three major populations (Figs. 3b-d), named ring density (RD)1, RD2 and RD3, were manually picked and aligned separately (Extended Data Fig. 4). We applied 9-fold symmetry during alignment and average, yielding three ring-like maps with diameters of approximately 50 nm, 65 nm and 100 nm for RD1, RD2,

and RD3, respectively (Fig. 3a), which appear in consistency with their original densities in the raw tomograms (Figs. 3b-d).” (Page 14, Line 16-23)

Comment 7: (4) *The fiducial beads free is the one advantage that the authors claimed in the paper, while fiducial beads are good reference especially when correlating the fluorescent image to cryoEM and help localize the region of interesting. Besides I haven't seen any statement of how the ROI is localized in cryoEM, if it is manual then it will be a big limitation for the application of this method to achieve high resolution, especially for small particles.*

Response 7:

We would like to thank this reviewer for his/her kind suggestions for our work. Our experience indicates that the idea of fiducial marker free can simplify the workflow of cryo-CLEM, increase the efficiency of specimen preparation, and improve the success ratio of cryo-FIB targeting. In addition, as we mentioned in our response #4, the effect of cryo-FIB milling induced distortion could attenuate the success ratio of cryo-FIB targeting for the conventional cryo-CLEM workflow even if the correlation was performed accurately based on the well distributed fiducial beads. Therefore, in our opinion, the fiducial beads based cryo-CLEM workflow for target cryo-lamella fabrication has its intrinsic limitation for wide application.

In the current study, since the size of centriole is large enough, we could easily find it directly from cryoEM image in low magnification (see Extended Data Figure 9 and Supplementary Dataset 2) and therefore we did not need the fluorescent image to help localize. However, we agree with this reviewer that the fiducial beads are useful when correlating the fluorescent image to cryoEM image for ROI localization and subsequent cryo-ET data collection.

Indeed, our ELI-TriScope based cryo-CLEM workflow (cryo-FM/ELI-TriScope/cryo-ET) is compatible with the usage of fiducial beads. In another separated manuscript of us (see bioRxiv 2022.09.16.508243; doi: <https://doi.org/10.1101/2022.09.16.508243>), we have developed another fiducial beads based cryo-CLEM workflow (HOPE-SIM cryoCLEM) by incorporating cryo-structure illumination microscopy (cryo-SIM) into our previous HOPE-cryoCLEM system (J Struct Biol. 2018;201(1):63-75). Besides, we also developed a 3D correlative software, 3D-View, to achieve accurate fiducial marker-based correlation between cryo-FM, cryo-FIB and cryo-EM. Therefore, by combining our HOPE-SIM and ELI-TriScope systems as well as the 3D-

View software, we could not only achieve a high success ratio of target cryo-lamella preparation, but also localize the region of interest accurately by correlating the cryo-SIM image with the cryoEM image based on the fiducial markers.

By the way, in the future, we could imagine we can invent a new cryo-electron microscope by introducing a cryo-fluorescence microscope in. The idea is like ELI-TriScope that we could calibrate the focal plane of FM to the object plane of TEM. For this kind of design, localization of region of interest is straightforward just like ELI-TriScope and there is no need to utilize the fiducial beads to help localize the region of interest.

As a result, we do not see any big limitation for the application of our ELI-TriScope solution to achieve high resolution even from smaller particles.

Decision Letter, first revision:

Dear Fei,

Thank you for submitting your revised manuscript "ELI trifocal microscope: A precise cryogenic fabrication system to prepare target cryo-lamellae of cells for in situ cryo-ET study" (NMETH-A49167A). It has now been seen by the original referees and their comments are below. The reviewers find that the paper has improved in revision, and therefore we'll be happy in principle to publish it in Nature Methods, pending minor revisions to satisfy the referees' final requests and to comply with our editorial and formatting guidelines.

Please be sure to clarify what you mean when you refer to your method as contactless, as requested by referee 2.

TRANSPARENT PEER REVIEW

Nature Methods offers a transparent peer review option for new original research manuscripts submitted from 17th February 2021. We encourage increased transparency in peer review by publishing the reviewer comments, author rebuttal letters and editorial decision letters if the authors agree. Such peer review material is made available as a supplementary peer review file. Please state in the cover letter 'I wish to participate in transparent peer review' if you want to opt in, or 'I do not wish to

participate in transparent peer review' if you don't. Failure to state your preference will result in delays in accepting your manuscript for publication.

Thank you again for your interest in Nature Methods Please do not hesitate to contact me if you have any questions.

Sincerely,
Rita

Rita Strack, Ph.D.
Senior Editor
Nature Methods

ORCID

Reviewer #3 (Remarks to the Author):

The response letter and the revised paper have clearly answered all my puzzle points. It is quite impressive that the authors presented all raw data to illustrate the true successful ratio of the experiment which is the key of CLEM research and obviously a breakthrough in this field. The author also remade Figure 3 to correct the previous misguiding of 9-fold symmetry map. With all correction and answer to my questions I am fully convinced by the result presented in the paper and look forward to it to be published on Nature Method.

Author Rebuttal, first revision:

Response to Reviewers

Reviewer #2:

I agree that the autogrid will typically protect the specimen during transfers from physical disruption by the tweezers (depending on the user). It is good that the workflow was developed to incorporate the autogrid. I still think it's unclear what is meant by "seamless and contactless" in the revised text.

If what you mean by "contactless" is that the workflow is completed entirely within an autogrid, it is simpler to say that directly, because it still implies that that autogrid assembly will need to be handled.

The term "seamless" in the context of cryo-transfers implies to me that everything is done under vacuum or nitrogen (gas or liquid), and not simply a minimized number of transfers through the air.

That being said, this is a great workflow that apparently works well. If I'm misunderstanding the workflow I apologize. I leave it to the editors to decide which wording is most appropriate.

Response:

We apologize for the ambiguity in the statements. To avoid any potential ambiguity as this reviewer suggested, we have deleted the words "contactless" / "seamless" and revised the statements in the manuscript as follows:

"Notably, the cryo-transfer between our HOPE system and the subsequent ELI-TriScope system **utilizes the AutoGrid to protect the EM grid**, minimizing the risk of specimen deformation, devitrification and ice contamination." (Page 9, Line 4-6).

And,

"Our cryo-transfer system **utilizes the AutoGrid to protect the EM grid** and the touch of the EM grid with the specimen can be largely avoided during the whole cryo-CLEM workflow." (Page 16, Line 18-21).

Final Decision Letter:

Dear Fei,

I am pleased to inform you that your Article, "ELI trifocal microscope: A precise cryogenic fabrication system to prepare target cryo-lamellae of cells for in situ cryo-ET study", has now been accepted for publication in Nature Methods. Your paper is tentatively scheduled for publication in our Feb or March print issue, and will be published online prior to that. The received and accepted dates will be May 6, 2022 and Dec 6, 2022. This note is intended to let you know what to expect from us over the next month or so, and to let you know where to address any further questions.

Over the next few weeks, your paper will be copyedited to ensure that it conforms to Nature Methods style. Once your paper is typeset, you will receive an email with a link to choose the appropriate publishing options for your paper and our Author Services team will be in touch regarding any additional information that may be required.

Your paper will now be copyedited to ensure that it conforms to Nature Methods style. Once proofs are generated, they will be sent to you electronically and you will be asked to send a corrected version within 24 hours. It is extremely important that you let us know now whether you will be difficult to contact over the next month. If this is the case, we ask that you send us the contact information (email, phone and fax) of someone who will be able to check the proofs and deal with any last-minute problems.

If, when you receive your proof, you cannot meet the deadline, please inform us at rjsproduction@springernature.com immediately.

Once your manuscript is typeset and you have completed the appropriate grant of rights, you will receive a link to your electronic proof via email with a request to make any corrections within 48 hours. If, when you receive your proof, you cannot meet this deadline, please inform us at rjsproduction@springernature.com immediately.

Once your paper has been scheduled for online publication, the Nature press office will be in touch to confirm the details.

Content is published online weekly on Mondays and Thursdays, and the embargo is set at 16:00 London time (GMT)/11:00 am US Eastern time (EST) on the day of publication. If you need to know the exact publication date or when the news embargo will be lifted, please contact our press office after you have submitted your proof corrections. Now is the time to inform your Public Relations or Press Office about your paper, as they might be interested in promoting its publication. This will allow them time to prepare an accurate and satisfactory press release. Include your manuscript tracking number NMETH-A49167B and the name of the journal, which they will need when they contact our office.

About one week before your paper is published online, we shall be distributing a press release to news organizations worldwide, which may include details of your work. We are happy for your institution or funding agency to prepare its own press release, but it must mention the embargo date and Nature Methods. Our Press Office will contact you closer to the time of publication, but if you or your Press Office have any inquiries in the meantime, please contact press@nature.com.

If you are active on Twitter, please e-mail me your and your coauthors' Twitter handles so that we may tag you when the paper is published.

Please note that *Nature Methods* is a Transformative Journal (TJ). Authors may publish their research with us through the traditional subscription access route or make their paper immediately open access through payment of an article-processing charge (APC). Authors will not be required to make a final decision about access to their article until it has been accepted. [Find out more about Transformative Journals](https://www.springernature.com/gp/open-research/transformative-journals)

To assist our authors in disseminating their research to the broader community, our SharedIt initiative provides you with a unique shareable link that will allow anyone (with or without a subscription) to read the published article. Recipients of the link with a subscription will also be able to download and print the PDF. As soon as your article is published, you will receive an automated email with your shareable link.

Please note that you and your coauthors may order reprints and single copies of the issue containing your article through Nature Portfolio 's reprint website, which is located at <http://www.nature.com/reprints/author-reprints.html>. If there are any questions about reprints please send an email to author-reprints@nature.com and someone will assist you.

Best regards,
Rita